# A system for functional studies of the major virulence factor of malaria parasites

Jakob Cronshagen[1,2], Johannes Allweier[2,3], Joëlle Paolo Mesén-Ramírez[1†], Jan Stäcker[1], Anna Viktoria Vaaben[4], Gala Ramón-Zamorano[1,5], Isabel Naranjo-Prado[1], Max Graser[1], Patricia López-Barona[1], Susann Ofori[2], Pascal WTC Jansen[6], Joëlle Hornebeck[1], Florian Kieferle[2], Agnes Murk[2], Elicia Martin[4], Carolina Castro-Peña[1], Richárd Bártfai[5], Thomas Lavstsen[4], Iris Bruchhaus[2,7]*, Tobias Spielmann[1]*

[1]Pathogen section, Bernhard Nocht Institute for Tropical Medicine, Hamburg, Germany; [2]Interface section, Bernhard Nocht Institute for Tropical Medicine, Hamburg, Germany; [3]Biophysics, Research Center Borstel, Leibniz Lung Center, Sülfeld, Germany; [4]Centre for translational Medicine & Parasitology, Department of Immunology and Microbiology, University of Copenhagen and Department of Infectious Diseases, Copenhagen, Denmark; [5]Department of Molecular Biology, Faculty of Science, Radboud University, Nijmegen, Netherlands; [6]Department of Molecular Biology, Radboud Institute for Molecular Life Sciences, Oncode Institute, Radboud University, Nijmegen, Netherlands; [7]Department of Biology, University of Hamburg, Hamburg, Germany

*For correspondence:
bruchhaus@bnitm.de (IB);
spielmann@bnitm.de (TS)

Present address: †Centre for Structural Systems Biology, Hamburg, Germany

## eLife Assessment

This study introduces an **important** approach using selection linked integration (SLI) to generate *Plasmodium falciparum* lines expressing single, specific surface adhesins PfEMP1 variants, enabling precise study of PfEMP1 trafficking, receptor binding, and cytoadhesion. By moving the system to different parasite strains and introducing an advanced SLI2 system for additional genomic edits, this work provides **compelling** evidence for an innovative and rigorous platform to explore PfEMP1 biology and identify novel proteins essential for malaria pathogenesis including immune evasion.

**Abstract** PfEMP1 is a variable antigen displayed on erythrocytes infected with the malaria parasite *Plasmodium falciparum*. PfEMP1 mediates binding of the infected cell to the endothelium of blood vessels, a cause of severe malaria. Each parasite encodes ~60 different PfEMP1 variants but only one is expressed at a time. Switching between variants underlies immune evasion in the host and variant-specific severity of disease. PfEMP1 is difficult to study due to expression heterogeneity between parasites which also renders genetic modification approaches ineffective. Here, we used selection-linked integration (SLI) to generate parasites all expressing the same PfEMP1 variant and genome edit the expressed locus. Moving this system from the reference strain 3D7 to IT4 resulted in PfEMP1 expressor parasites with effective receptor binding capacities. We also introduce a second version of SLI (SLI2) to introduce additional genome edits. Using these systems, we study PfEMP1 trafficking, generate cell lines binding to the most common endothelial receptors, survey the protein environment from functional PfEMP1 in the host cell, and identify new proteins needed for PfEMP1-mediated sequestration. These findings show the usefulness of the system to study the key virulence factor of malaria parasites.

## Introduction

A key factor for the pathology of the human malaria parasite *Plasmodium falciparum* is its capacity to render the infected red blood cells (RBCs) adherent to the endothelium of blood vessels (*Newbold et al., 1999*). This cytoadhesion allows the parasite to escape spleen-mediated clearance of infected RBCs (*Borst et al., 1995*) but causes sequestration of infected RBCs in major organs, which can lead to severe, life-threatening complications including cerebral malaria (*Miller et al., 2002*).

Cytoadhesion is mediated by members of the *P. falciparum* erythrocyte membrane protein 1 (PfEMP1) family. PfEMP1s are 150–450 kDa single-pass transmembrane proteins inserted into the membrane of the infected RBC (*Baruch et al., 1995*; *Gardner et al., 2002*; *Leech et al., 1984*; *Smith et al., 1995*). PfEMP1s are encoded by the two-exon *var* genes, with exon 1 encoding the variable extracellular part of PfEMP1 which has diversified to bind different host receptors such as CD36, ICAM-1, EPCR, and CSA through its DBL and CIDR domains (*Baruch et al., 1995*; *Smith et al., 1995*; *Kyes et al., 2007*; *Salanti et al., 2004*; *Salanti et al., 2003*; *Scherf et al., 1998*; *Smith, 2014*; *Turner et al., 2013*). The *var* exon 2 encodes a conserved intracellular C-terminal part, the acidic terminal segment (ATS), which anchors the PfEMP1 underneath the RBC membrane in so-called knobs, parasite-induced elevations of the RBC membrane which contribute to efficient cytoadhesion of the infected RBC (*Ruangjirachuporn et al., 1992*; *Crabb et al., 1997*; *Sanchez et al., 2019*). Each parasite genome contains ~45–90 *var* genes that differ in sequence within and between parasites, but confers each parasite a similar repertoire of human receptor-binding phenotypes (*Gardner et al., 2002*; *Otto et al., 2019*; *Rask et al., 2010*; *Thompson et al., 1997*). Each parasite expresses only one *var* gene at a given time but can switch to a different *var* gene, resulting in antigenic variation (*Scherf et al., 1998*; *Chen et al., 1998*; *Voss et al., 2014*). While the diversity of *var* genes between isolates is high, the unique VAR2CSA PfEMP1 binding placental CSA - the cause of the detrimental sequestration leading to pregnancy malaria - is much more conserved between different isolates (*Salanti et al., 2004*; *Salanti et al., 2003*).

PfEMP1 is the major target for the protective acquired immune response (*Chan et al., 2012*) and *var* gene switching is important to escape immune recognition and a mechanism to establish long-term infection in the host (*Voss et al., 2014*; *Chan et al., 2012*; *Bull et al., 1998*; *Bull and Marsh, 2002*; *Guizetti and Scherf, 2013*; *Kyes et al., 2001*; *Wichers-Misterek et al., 2023*). Specific PfEMP1 variants are associated with pathology in the human host and with its immune status (*Salanti et al., 2004*; *Turner et al., 2013*; *Smith et al., 2013*; *Wichers et al., 2021*). Understanding the binding properties of individual PfEMP1 variants, antibody recognition, and switching is therefore critical to understand the pathology of malaria.

How PfEMP1 reaches its final destination at the host cell membrane is only partially understood. Exported parasite proteins are translocated by the PTEX complex into the host cell, but it is not fully clear if this is also true for PfEMP1 (*Batinovic et al., 2017*; *Beck et al., 2014*; *de Koning-Ward et al., 2009*; *Elsworth et al., 2014*; *McMillan et al., 2013*; *Riglar et al., 2013*). Once in the host cell, PfEMP1 is most abundantly found at parasite-induced vesicular cisternae termed Maurer's clefts, and only a small fraction of all PfEMP1 molecules reach the host cell surface (*Sanchez et al., 2019*). How PfEMP1 is transported within the host cell to reach the surface is unclear, but a number of other exported proteins, for example SBP1 and PTP1-7, are needed for that process (*Carmo et al., 2022*; *Cooke et al., 2006*; *Maier et al., 2007*; *Rug et al., 2014*).

A key problem in studying PfEMP1 lies in the heterogeneous *var* gene expression of the parasites in cell culture. This results in a mixed population of cells that have different antigenic and binding properties. Selective enrichment of binding phenotypes through elaborate panning of parasites against receptors or antibodies (*Avril et al., 2012*; *Claessens et al., 2012*; *Cooke et al., 1996*; *Nunes-Silva et al., 2015*) or the utilization of parasite strains with more stable PfEMP1 expression, such as CS2 (*Cooke et al., 1996*; *Maier et al., 2008*), has previously been used to circumvent this issue. A further problem is that specific PfEMP1s can be difficult to detect at the protein level. Antibodies against the conserved ATS do not distinguish between PfEMP1 variants and often cross-react with RBC spectrin (*Nilsson et al., 2012*). Extracellular domain-specific antibodies need to be generated for each newly studied PfEMP1 (*Smith et al., 1995*). Furthermore, the large size hampers episomal expression, and in some cases, episomally expressed mini-PfEMP1s were used as a surrogate, for example to study PfEMP1 trafficking (*Batinovic et al., 2017*; *McMillan et al., 2013*; *Looker et al., 2019*; *Melcher et al., 2010*). Finally, research questions needing genetic

modification of PfEMP1s pose the problem that the modified locus is only expressed in some of the parasites.

Here, we use selection-linked integration (SLI; *Birnbaum et al., 2017*) to generate parasite lines that each predominantly express one specific PfEMP1 (*Omelianczyk et al., 2020*). This permitted us to generate different parasite lines with binding specificities against all major binding receptors and parasites with modified PfEMP1s. We also introduce SLI version 2 (SLI2) to obtain a second genomic integration in parasites that already have a SLI-based alteration to express a specific tagged PfEMP1. We show that our approach can be used to study mutually exclusive expression of *var* genes, track the activated PfEMP1 via a small tag, study its trafficking, endothelial receptor binding, its proxiome in living parasites, and identify novel proteins needed for PfEMP1-mediated cytoadhesion.

## Results
### Activation of specific PfEMP1 in the total cell population in 3D7 parasites

Using SLI (*Birnbaum et al., 2017*), parasites were genetically modified to be resistant to G418 if expressing a targeted *var* gene (*Figure 1A*), permitting selection of a population of parasites expressing the desired PfEMP1. In addition, the chosen PfEMP1 obtains a C-terminal 3xHA tag to specifically detect it (*Figure 1A*). We first aimed to generate two different 3D7 parasite lines: in the first, we targeted PF3D7_0809100 (*3D7var0809100*, the predominant *var* gene in our 3D7 wildtype *Figure 1—figure supplement 1A*), and in the second, PF3D7_1200600, the 3D7 VAR2CSA-encoding *var* gene. Cell lines with the expected genomic modification were obtained in both cases (3D7var0809100-HA[endo] and 3D7var2csa-HA[endo] parasites) and the HA-tagged PfEMP1 was detected at the Maurer's clefts in the host cell (*Figure 1B and C*), indicating that the cells in the culture expressed the desired PfEMP1 and that it could conveniently be detected via the epitope tag. qPCR showed predominant expression of the activated *var* genes (*Figure 1E and F*). This was confirmed by RNA-Seq (*Figure 1—figure supplement 1B*, *Source data 1*), which showed high read coverage across the desired *var* gene, whereas transcripts of all other *var* genes were negligible (example in *Figure 1—figure supplement 1C*, *Source data 1*). As PfEMP1 surface exposure is not typically detected using standard immunofluorescence assays, we conducted trypsin digestion assays with intact infected RBCs (*Waterkeyn et al., 2000*) which showed a protected fragment indicative of surface exposure of the HA-tagged PfEMP1 (*Figure 1D*). When we lifted G418 pressure for 3 weeks, the dominance of the targeted PfEMP1 declined in favor of a more heterogeneous *var* expression profile, indicating that switching to other *var*s was still possible in these parasites (*Figure 1E and F*).

Previous work indicated that SLI to select parasites expressing a specific *var* gene can influence transcription of neighboring genes that are oriented head-to-tail (*Omelianczyk et al., 2020*). We did not observe activation of head-to-tail oriented *rif*s in our 3D7var0809100-HA[endo] and 3D7var2csa-HA[endo] SLI lines, as the corresponding *rif*s showed no or negligible transcription in RNA-Seq (*Figure 1—figure supplement 1D*). To further look into co-activation, we used SLI to select for parasites expressing a *var* gene that shares a promoter region with a *rif* gene in a head-to-head orientation (PF3D7_0425800: cell line 3D7var0425800-HA[endo]) (*Figure 1G*, *Figure 1—figure supplement 1E*). Correct integration of the plasmid into the genome and expression of the tagged PfEMP1 was confirmed (*Figure 1G*). RNA-Seq showed predominant expression of the activated *var* gene (*Figure 1G*) but also transcription of the neighboring *rif* gene (Figure S1F) with ~40% of all *rif* transcripts belonging to this *rif* gene (PF3D7_0425700; *Figure 1—figure supplement 1G*).

We also activated two *rif* genes (PF3D7_0425700: cell line 3D7rif0425700-HA[endo] and PF3D7_1254800: cell line 3D7rif1254800-HA[endo]; *Figure 1—figure supplement 1H*). The two resulting cell lines had the correct genomic modification and IFAs indicated expression of the HA-tagged RIFINs (*Figure 1—figure supplement 1H*). qPCR showed that in both lines the activated *rif* gene was the most expressed (~65% and ~40% of *rif* transcripts; *Figure 1—figure supplement 1I*). While in the 3D7rif1254800-HA[endo] parasite line, the *var* expression profile looked similar to the 3D7 parent with predominant expression of *var* PF3D7_0809100 located on a different chromosome (*Figure 1—figure supplement 1I*), activation of *3D7rif0425700* (which, in contrast to *3D7rif1254800,* has a *var* gene in head-to-head orientation) led to co-activation of the neighboring *var* gene (PF3D7_0425800; *Figure 1—figure supplement 1I*). Overall, the data with activated *var*s and *rif*s suggests that

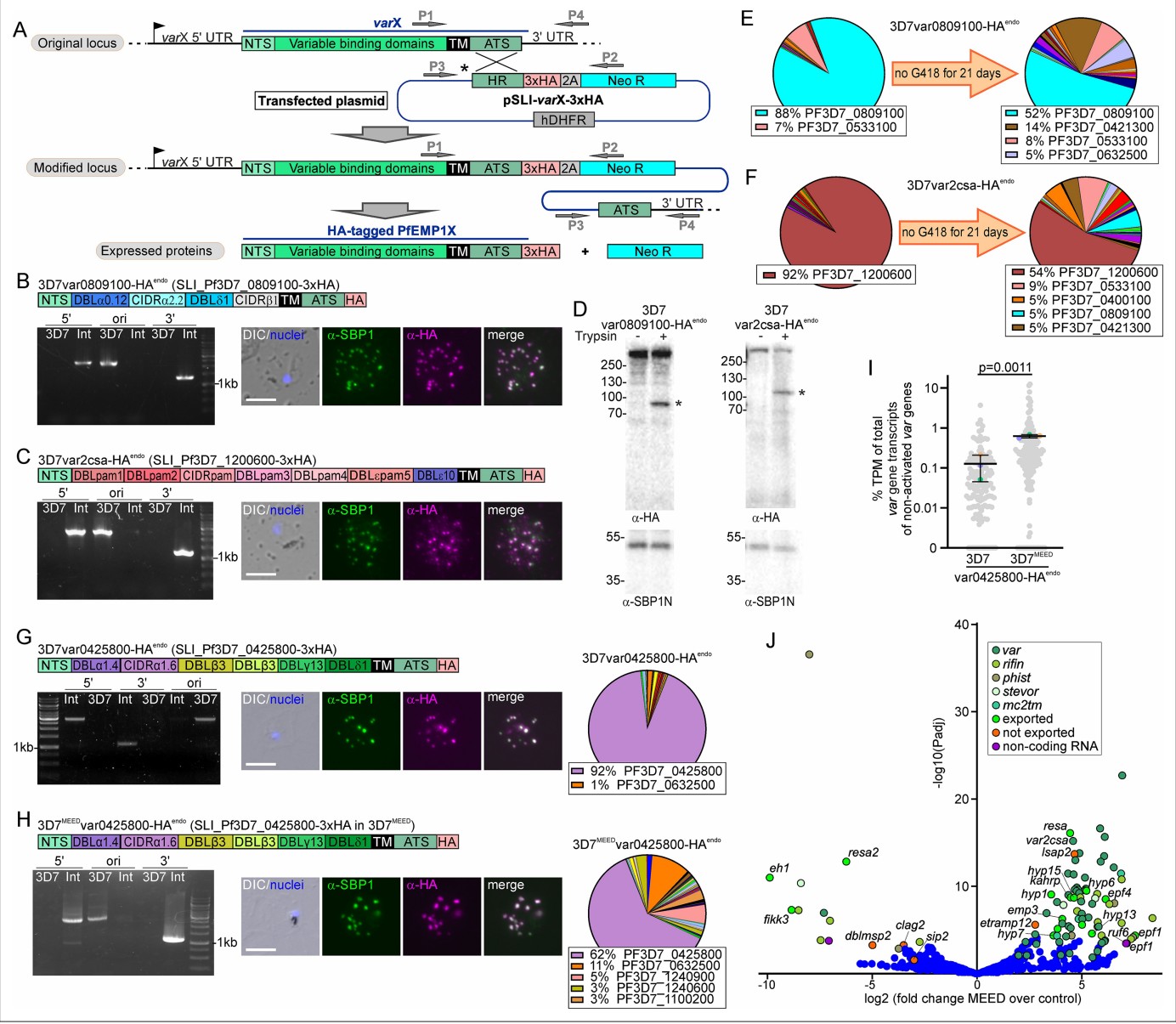

**Figure 1.** SLI-activation of *var* genes in 3D7. (**A**) Schematic for SLI strategy. HR: homology region; ATS: acidic terminal segment; NTS: N-terminal segment; 2 A: T2A skip peptide; NEO-R: G418-resistance; hDHFR: human dihydrofolate reductase; arrows P1-4: primers for diagnostic PCR; X: desired *var* gene. (**B, C**) Activation of indicated PfEMP1. Scheme shows domain organization. Agarose gels show PCR products confirming correct integration of the SLI plasmid. Product over 5′ integration junction (5′): P1+P2; 3′ integration junction (3′): P3+P4; original locus (ori): P1+P4; see (**A**) for primer positions, see *Table 1* for sequence of primers used; 3D7: parent; Int: integrant cell line. Fluorescence microscopy images show IFAs with indicated antibodies. Nuclei: Hoechst 33342; DIC: differential interference contrast; size bars 5 µm. (**D**) Western blot of trypsin cleavage assays with indicated parasites. Asterisks show protected PfEMP1 fragment. α-SBP1-N: control for integrity of host cell (breach of RBC membrane would result in a smaller SBP1 fragment). Marker in kDa. (**E, F**) Pie charts with proportions of total *var* gene transcripts determined by qPCR of the indicated cell lines on G418 and after lifting G418. (**G, H**) Activation of PF3D7_0425800 in 3D7 or 3D7^MEED. Scheme shows domain organization. Agarose gels show PCR products confirming correct integration of the SLI plasmid as described in (**A**). Fluorescence microscopy images show IFAs as described in (**B, C**). Pie charts show proportions of total *var* gene transcripts of the indicated cell lines determined by RNAseq (normalized to TPM). (**I**) SuperPlot (***Lord et al., 2020***) showing percentage (log scale) of total *var* gene transcripts for non-activated *var* genes of the indicated cell line determined by RNAseq (normalized to TPM; small gray dots: individual *var* genes; large colored dots: average of each replicate; bars: mean of averages of replicates with SD; n=3 biological replicates; unpaired t-test; p-values indicated). See also ***Source data 1***. (**J**) Volcano plot showing differential expression (RNASeq) of 3D7 or 3D7^MEED both containing the same SLI modification to express PF3D7_0425800. Selected hits were color-coded as indicated. 'Exported' refers to all proteins that are known or predicted to be exported but do not fall into the selected families of exported proteins labeled with other colors. Short names are given for color-coded hits when available (full names, accession, and total data in ***Figure 1—source data 4***).

*Figure 1 continued on next page*

*Figure 1 continued*

The online version of this article includes the following source data and figure supplement(s) for figure 1:

**Source data 1.** qPCR corresponding to panels E and F.

**Source data 2.** Unedited agarose gels shown in panels B, C, G, and H.

**Source data 3.** Agarose gels shown in panels B, C, G, and H with annotation.

**Source data 4.** Full and unedited blots corresponding to panel D.

**Source data 5.** Full and unedited blots annotated and indicating the regions shown in panel D.

**Source data 6.** RNASeq data of 3D7 vs 3D7$^{MEED}$ corresponding to panel J.

**Figure supplement 1.** Expression of desired and co-activated genes.

**Figure supplement 1—source data 1.** qPCR corresponding to panels A and E.

**Figure supplement 1—source data 2.** Unedited agarose gels shown in panel H.

**Figure supplement 1—source data 3.** Unedited agarose gels shown in panel H with annotations.

**Figure supplement 1—source data 4.** qPCR corresponding to panel I.

neighboring genes can be co-activated if in head-to-head orientation, likely due to a shared promoter region affected by the epigenetic changes resulting in the expression of the SLI-targeted *var* gene. While SLI-activation of *rif* genes also led to the dominant expression of the targeted *rif* gene, other *rif* genes still took up a substantial proportion of all detected *rif* transcripts, speaking against a mutually exclusive expression in the manner seen with *var* genes.

## Validation of a parasite line with impaired mutually exclusive *var* expression

Previous work described a 3D7 line expressing multiple *var* genes in single infected RBCs (*Joergensen et al., 2010*), likely due to a defective *var* regulation system, here designated 3D7$^{MEED}$ (for 'mutually exclusive expression defective'). We used SLI to obtain parasites expressing PF3D7_0425800 in the 3D7$^{MEED}$ parasites (3D7$^{MEED}$var0425800-HA$^{endo}$; *Figure 1H*). In contrast to standard 3D7 with the same SLI modification to express *3D7var0425800*, 3D7$^{MEED}$ showed elevated levels of transcription of multiple *var* genes in addition to the activated one, both by qPCR (*Figure 1—figure supplement 1E*) and by RNA-Seq (*Figure 1H and I*). Anti-HA IFAs showed that the 3D7$^{MEED}$ parasite nevertheless expressed the activated PfEMP1 (all trophozoites were HA-positive in IFAs (n=82 parasites from four independent experiments)) (*Figure 1H*), indicating that individual parasites expressed multiple *var* genes. This line might therefore be an interesting tool to study mutually exclusive expression, silencing, and switching mechanisms.

In an attempt to find changes that may cause the MEED phenotype, we compared all differentially expressed transcripts (161 down- and 93 up-regulated using a log2 fold change of >2 and Padjusted of <0.05 as cut off) of the 3D7 vs 3D7$^{MEED}$ parasites (*Figure 1J*, *Figure 1—source data 6*). This confirmed the upregulation of most *var* genes in the 3D7$^{MEED}$ parasites. Members of some other gene families encoding exported proteins and genes of other exported proteins were also upregulated. Many of these may be co-regulated with the *var*s, as for instance many of the upregulated *rif* gene loci are close to upregulated *var* loci (*Figure 1—source data 6*). However, a few genes encoding exported proteins were also downregulated. Concentrating on non-exported proteins to identify potential changes responsible for the MEED phenotype, we noticed that the transcripts encoding the ApiAP2 protein SIP2 (*Flueck et al., 2010*) were down ~eightfold (pAdjusted ~0.025; *Figure 1J*, *Figure 1—source data 6*). SIP2 was previously shown to bind heterochromatin in subtelomeric and telomeric regions, including certain *var* promoters (*Flueck et al., 2010*). Its downregulation might result in changes to chromosome end biology influencing *var* silencing. The other potentially causal change was an upregulation of the non-coding RNA *ruf6* for which overexpression has been shown to impair monoallelic *var* gene expression (*Guizetti et al., 2016*). While both *sip2* downregulation and *ruf6* upregulation are possible explanations for the relaxed silencing of *var* genes in the 3D7$^{MEED}$ parasites, independent experiments are needed to confirm that any of these changes are reasons for the MEED phenotype.

**Table 1.** Primers for confirmation of integration of plasmids into the genome.

| Name of primer/target | Direction | Sequence |
| --- | --- | --- |
| *Plasmid region primers (P2, P3, P7, P8)* | | |
| Neo40 (P2) | rv | CGAATAGCCTCTCCACCCAAG |
| pARL55 (P3/P7) | fw | GGAATTGTGAGCGGATAACAATTTCACACAGG |
| GFP85 (P8) | rv | ACCTTCACCCTCTCCACTGAC |
| Ty1 (P8) | rv | GTGGATCTTGATTTGTATGC |
| | | |
| *Gene-specific primers (P1, P4, P5, P6)* | | |
| 3D7var0809100-HAendo | fw | CCCCCAGTTCCTGCTCCAGCTGGTG |
| | rv | CCTAATGCATATTATGAAATATCCAC |
| 3D7var2csa-HAendo | fw | GGTGGGACATGAATAAATATCACATATGGG |
| | rv | CTTTCCATATATTTTATGCATTGCATTTATTAG |
| 3D7var0425800-HAendo | fw | GTAGATGAATGGATAAAGCTGAAAAAGG |
| | rv | CAAAAAATTATGAATCGAATATATTTAG |
| 3D7MEEDvar0425800-HAendo | fw | GTAGATGAATGGATAAAGCTGAAAAAGG |
| | rv | CAAAAAATTATGAATCGAATATATTTAG |
| 3D7rif0425700-HAendo | fw | GTTTTATGTTAAACATATTTGATGTATTTATAAC |
| | rv | GCGCAAAATAATTCATTCATTAAAATACCTG |
| 3D7rif1254800-HAendo | fw | GTTATAGTTTTTATCATAAAATAATATACGTATCAC |
| | rv | CAGTACATGTACCAAACATCCTACCAACATCTAC |
| 3D7var0809100-mDHFR-HAendo | fw | CCCCCAGTTCCTGCTCCAGCTGGTG |
| | rv | CCTAATGCATATTATGAAATATCCAC |
| 3D7var2csa-mDHFR-HAendo | fw | GGTGGGACATGAATAAATATCACATATGGG |
| | rv | CTTTCCATATATTTTATGCATTGCATTTATTAG |
| IT4var66-HAendo | fw | TAATATGAGTACTAATAGTATGG |
| | rv | AAACTCCACATAAAAAAATAAAAATCAAAC |
| IT4var2csa-HAendo | fw | TAGATATATCCCCTATGTGAGTGATAC |
| | rv | ATATACACATATAAATCATCACC |
| IT4var01-HAendo | fw | CGACAACCACGTGAAGTGACGCATTCCATAGTC |
| | rv | CTAATATAGTATCCATAGTAGAATTATCAGG |
| IT4var16-HAendo | fw | AGTCCTAAATATAAAACATTGATAGAAGTGG |
| | rv | AATAAAAAGAAATAATAATATATCG |
| IT4var19-HAendo | fw | ACATTGATAGAAGTGGTACTAGAACCATCG |
| | rv | AAAAAATTCAAACATATGTATATACATACG |
| PTP1-TGD | fw | TAGAATAACATATAAAAAATATGTATTCTG |
| | rv | TTTAACTTTACAAATTCCTTTTAATTTACG |
| IT4var01-BirA*Pos1endo | fw | CGACAACCACGTGAAGTGACGCATTCCATAGTC |
| | rv | CTAATATAGTATCCATAGTAGAATTATCAGG |
| IT4var01-BirA*Pos2endo | fw | TTAAGGATGATTGTCGTAGTGACACCCCAG |
| | rv | CTAATATAGTATCCATAGTAGAATTATCAGG |

*Table 1 continued on next page*

*Table 1 continued*

| Name of primer/target | Direction | Sequence |
|---|---|---|
| IT4var01-BirA*Pos3endo | fw | TTAAGGATGATTGTCGTAGTGACACCCCAG |
| | rv | AGGTATTCCATAATCTCCTTTAGGTATATCAATAACAC |
| TryThrA-Ty1 | fw | TTGTTTTTGTCGTATAACAGAACCAATGG |
| | rv | GTACATAACAAAAATGGTATATTAAAAAGC |
| TryThrA-TGD | fw | TTGTTTTTGTCGTATAACAGAACCAATGG |
| | rv | CATTAGACATTCCAGAATTTTCATATTTTTCC |
| PTEF-Ty1 | fw | GAAAATGAAAGATGATGACTATGATGAAAG |
| | rv | ACAAAAAAACAAAACAAAATTTTGATTAGG |
| PTEF-TGD | fw | GGTTCTATTTTTATATAAGTAATCACATAC |
| | rv | ATAATAATCTGTTTCATCAATATCATGTTC |
| EMPIC3-Ty1 | fw | AAAAAGTATGAATTATTTGGTGTGAACAAG |
| | rv | TATCTAATTGCATATAAAATTTTACAACAG |
| EMPIC3-TGD | fw | AAAAAGTATGAATTATTTGGTGTGAACAAG |
| | rv | TATCTAATTGCATATAAAATTTTACAACAG |
| PeMP2-Ty1 | fw | AATTCAAGAATATAATTCAATTAGTTCTTC |
| | rv | TTATTTCATTTACGAAAACACCATTTTCAC |
| PeMP2-TGD | fw | AATTCAAGAATATAATTCAATTAGTTCTTC |
| | rv | GTTCCTTATGTATTGATCTTCTTGCTCTGC |
| PTP7-TGD | fw | ATGGTTTTATTTATTTTTCAATGGAAAAAG |
| | rv | CATAATTTTCCTCATCTTCACTATTCTCCG |

## Transport of PfEMP1 into the host cell

Next, we tested if SLI would permit obtaining parasites all expressing a modified PfEMP1 to track and study its transport. Limited overlap of PfEMP1 with PTEX components had raised the question whether it is transported via PTEX or not (*McMillan et al., 2013*; *Riglar et al., 2013*). While ablating PTEX function blocks PfEMP1 transport, indicating PTEX-dependent PfEMP1 transport (*Beck et al., 2014*; *Elsworth et al., 2014*), this may also be an indirect effect as inhibiting PTEX function also blocks the transport of other exported proteins essential for PfEMP1 transport. To directly assess PfEMP1 transport through PTEX, we used SLI to obtain parasites expressing VAR-0809100 (PF3D7_0809100) or VAR2CSA (PF3D7_1200600) and at the same time tagged them with mDHFR-3xHA (*Figure 2A*, *Figure 2—figure supplement 1*). The folding of the mDHFR domain can be stabilized by addition of WR99210 (WR) which prevents transport through translocons requiring unfolding (*Eilers and Schatz, 1986*) and can be used to assess PTEX passage of soluble and transmembrane proteins in *P. falciparum* parasites (*Gehde et al., 2009*; *Grüring et al., 2012*; *Figure 2A*). Both mDHFR-fused PfEMP1s were efficiently exported to the Maurer's clefts but not blocked when WR was added (*Figure 2A*). While this suggests that PfEMP1 might not be transported via PTEX, we previously noted that the length of the region between the transmembrane and mDHFR domain influences its blocking properties in exported transmembrane proteins (*Mesén-Ramírez et al., 2016*) and it therefore cannot be fully excluded that this protein still uses PTEX for transport. To circumvent this problem, we exploited the property of proteins that – when fused to mDHFR – can be conditionally arrested in the PTEX translocon, preventing the passage of other exported proteins (*Mesén-Ramírez et al., 2016*). Blocking PTEX in that manner (using SBP1-mDHFR-GFP conditionally arrested in PTEX) prevented PfEMP1 transport (*Figure 2B*), suggesting the need of PTEX function for PfEMP1 transport. However, similarly to previous work inactivating PTEX components (*Beck et al., 2014*; *Elsworth et al., 2014*), this does not exclude that PfEMP1 trafficking was prevented due to other exported proteins needed for

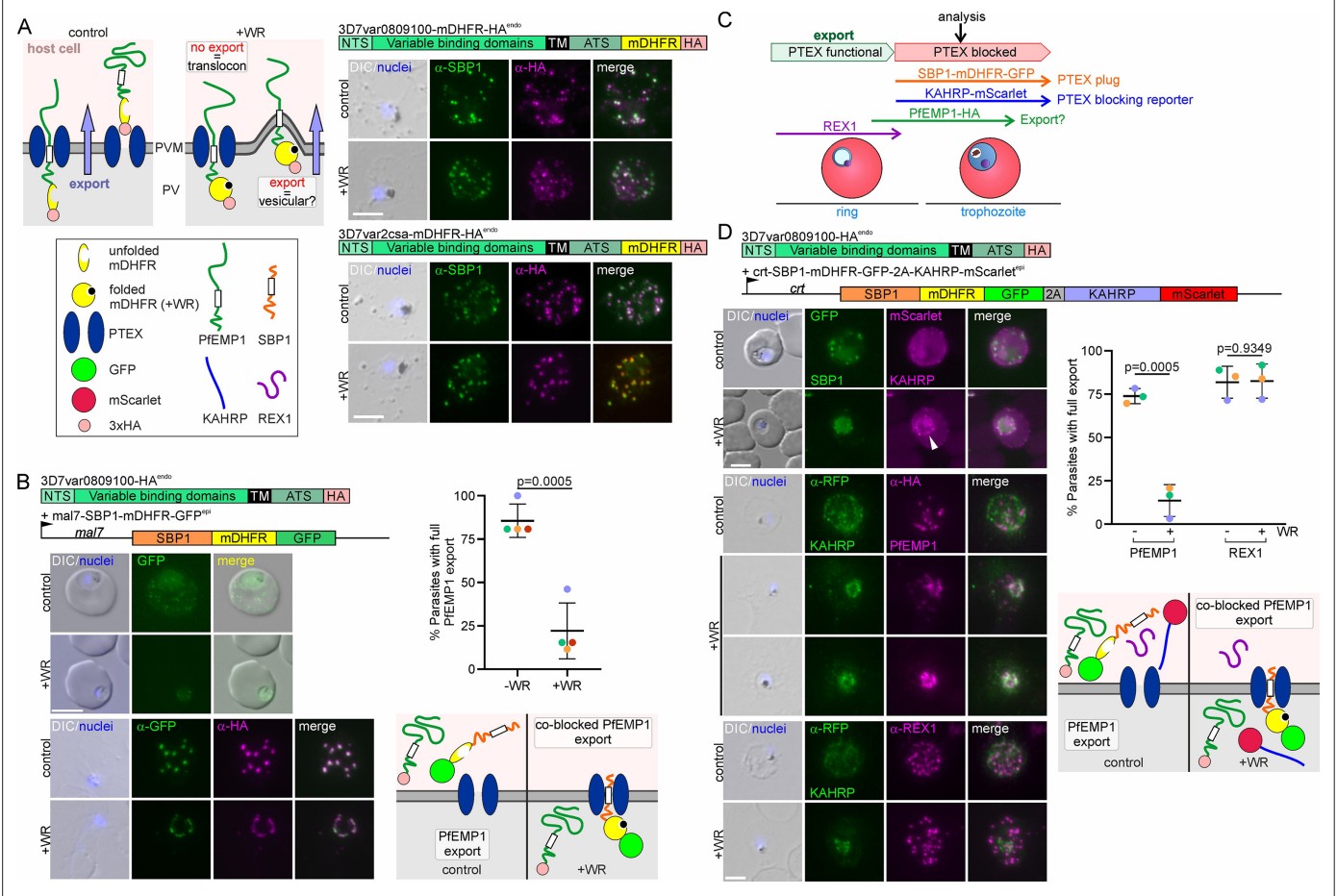

**Figure 2.** Clogging PTEX prevents PfEMP1 transfer into the host cell. (**A**) Scheme: options for impact of WR-induced stabilization of mDHFR folding on PfEMP1 export. Relevant domains of modified PfEMP1 indicated. Fluorescence microscopy images of IFAs with parasites of the indicated cell line + and –WR with the indicated antibodies. Nuclei: Hoechst 33342; DIC: differential interference contrast; size bars 5 µm. (**B**) Effect of blocking PTEX (+WR) with early (*mal7* promoter) expressed SBP1-mDHFR-GFP on PfEMP1 export. Relevant expressed products are shown. Live cell images (top rows) and IFAs (bottom rows; as described in (**A**)) of parasites grown + and -WR. Graph: quantification of parasites with a PfEMP1 export phenotype + and -WR (four biological replicates; dots: % cells per replicate; bars: mean of replicates with SD; n=26 parasites per experiment and condition; +WR only parasites with an SBP1-mDHFR-GFP export phenotype were scored; unpaired t-test; p-values indicated). Scheme shows WR-dependent clogging of PTEX (right) or control (left); features explained in (**A**). (**C**) Effect of blocking PTEX with late (*crt* promoter) expressed SBP1-mDHFR-GFP-2A-KAHRP-mScarlet on PfEMP1 export. Relevant expressed products are shown. Live cell images (top rows) and IFAs (bottom rows, as described in **A**) + and -WR. Graph: quantification of parasites with PfEMP1 or REX1 export phenotype + and -WR (3 biological replicates; -WR, PfEMP1: n=34, 76, 60; +WR, PfEMP1: n=18, 48, 30; -WR, REX1: n=18, 27, 35; +WR, REX1: n=12, 31, 25), only parasites with a KAHRP-mScarlet (late PTEX block reporter) export phenotype were scored (dots: % cells per replicate; bars: mean of replicates with SD; unpaired t-test; p-values indicated). The scheme shows WR-dependent clogging of PTEX (right) or control (left); features explained in (**A**); note that due to late block, early expressed REX1 is in the host cell in both conditions.

The online version of this article includes the following source data and figure supplement(s) for figure 2:

**Source data 1.** PfEMP1 export blocked by SBP1mDHFR, corresponding to panel B.

**Source data 2.** PfEMP1 and REX1 export block, corresponding to panel D.

**Figure supplement 1.** Confirmation of correct integration for PfEMP1 mDHFR fusion parasites.

**Figure supplement 1—source data 1.** Unedited agarose gels are shown in the figure.

**Figure supplement 1—source data 2.** Unedited agarose gels are shown in the figure with annotations.

PfEMP1 that were themselves prevented from export through PTEX. We therefore expressed the PTEX blocking construct later in the cycle (using the *crt* promoter) in an attempt to block PTEX passage only after the PfEMP1 trafficking proteins had already reached the host cell (*Figure 2C*). Using this strategy, the early expressed PfEMP1-transport protein REX1 was still exported while a later-stage episomally expressed mScarlet-tagged KAHRP reporter could be blocked and was used to monitor clogging of

PTEX after the parasite ring stage (*Figure 2C and D*). We then inspected REX1 and PfEMP1 transport in the cells where the KAHRP-mScarlet reporter showed a late block of PTEX (*Figure 2D*). REX1 was exported in most of the cells with a blocked KAHRP reporter, indicating that proteins needed for PfEMP1 trafficking were not hindered in reaching the host cell (*Figure 2D*). However, PfEMP1, which is later expressed, showed accumulation around the parasite in the majority of cells (*Figure 2C*). This indicated that clogging PTEX later in the cycle prevented PfEMP1 transport, supporting that PfEMP1 passes through PTEX. While we monitored only REX1, most PfEMP1-trafficking proteins show a similar early expression (*McMillan et al., 2013*; *Grüring et al., 2011*; *Marti et al., 2004*), indicating the effect may be direct, favoring the idea that PfEMP1 passes through PTEX. This would also mean that the mDHFR-based translocation block is not effective in the PfEMP1-mDHFR fusion construct.

## SLI PfEMP1 expressor cell lines for binding studies using IT4 parasite strain

Next, we tested whether SLI PfEMP1-expressor parasites are useful for PfEMP1 binding studies. Initial experiments using 3D7 parasites with activated *vars* showed no or only minimal binding of infected RBCs to receptors (see below). We therefore moved to the FCR3/IT4 clone generally considered a cytoadhesion-competent parasite line (*Bourke et al., 1996*; *Udeinya et al., 1983*) and used SLI to generate parasites expressing PfIT_040025500 (*IT4var66*), predicted to encode a CD36-binding PfEMP1 (*Hsieh et al., 2016*) with a similar domain composition to 3D7var080910, as well as PfIT_120006100 (*IT4var2csa*), encoding the IT4 VAR2CSA variant (*Figure 3A and B*). Also, in IT4, SLI was effective to obtain parasites expressing the targeted *vars*, and based on IFAs, the HA-tagged PfEMP1 was expressed in the corresponding parasite lines (*Figure 3A and B*). RNA-Seq showed predominant expression of the activated *var* genes (*Figure 3C* and *Source data 1*), and trypsin assays that the expressed PfEMP1 was presented on the RBC surface (*Figure 3D*). For binding studies, we developed and validated a semi-automated pipeline to score the number of bound infected RBCs in binding assays to permit the unbiased and higher throughput scoring of bound infected RBCs and increase the number of fields that can be analyzed per assay (*Figure 3—figure supplement 1*). Both IT4 lines showed the expected receptor binding. The IT4var2csa-HA[endo] infected RBCs bound both decorin-coated slides and endothelial cells expressing CSA (HBEC-5i; *Dörpinghaus et al., 2020*), and binding was inhibited by soluble CSA (*Figure 3E and F*). RBCs infected with IT4var66-HA[endo] bound to CHO-CD36 but not CHO-GFP or CHO-ICAM-1 cells (*Figure 3G*). In contrast, the 3D7 parasites expressing VAR2CSA (3D7var2csa-HA[endo]) or 3D7var0809100 (3D7var0809100-HA[endo], predicted to bind CD36 *Hsieh et al., 2016*) did not or only poorly bind their respective receptors (*Figure 3E–G*) despite the expressed PfEMP1 being detectable on the surface (*Figure 1D*). While the binding properties of PfEMP1 are difficult to compare between strains, VAR2CSA was chosen because it is well-conserved between isolates, permitting a comparison of binding efficiencies between 3D7 and IT4. Hence, these findings indicate that IT4 is a better cytoadhesion binder than the 3D7 used in our lab.

VAR2CSA is assumed to be the only PfEMP1 encoded in the genome that binds CSA. No binding to CSA was observed with the parental IT4 parasites (*Figure 3E and F*) in agreement with the low levels of *var2csa* transcripts in these parasites (*Figure 1—figure supplement 1J*). This indicated that SLI targeting *var2csa* selected this binding phenotype from undetectable levels. Overall, we conclude that SLI generated PfEMP1 expressor lines in IT4 can be used to study binding of specific PfEMP1 and that 3D7 - at least the version from our lab - is less suitable.

## IT4 parasites with further binding properties

In order to extend the repertoire of cell lines to study PfEMP1 binding, we selected two known or suspected CD36- and ICAM-1-binders (PfIT_060021400; cell line IT4var01-HA[endo] and PfIT_120024500; cell line IT4var16-HA[endo]; *Howell et al., 2008*; *Janes et al., 2011*; *Metwally et al., 2017*) and an EPCR-binder (PfIT_010005000: cell line IT4var19-HA[endo]; *Turner et al., 2013*). Correct generation of the cell lines and expression of the desired *var* gene was confirmed (*Figure 4A–C*). Trypsin assays showed surface exposure of the *IT4var01* and *IT4var16* but not *IT4var19* encoded PfEMP1 (*Figure 4D*). Both RBCs infected with IT4var01-HA[endo] and IT4var16-HA[endo] bound to CHO-CD36 and CHO-ICAM-1 cells but not GFP-expressing CHO cells, in agreement with the expected binding properties (*Figure 4E*). IT4var16-HA[endo] parasites showed a higher binding capacity to ICAM-1 than IT4var01-HA[endo] parasites, whereas IT4var01-HA[endo] parasites showed proportionally more binding to CD36 than IT4var16-HA[endo]

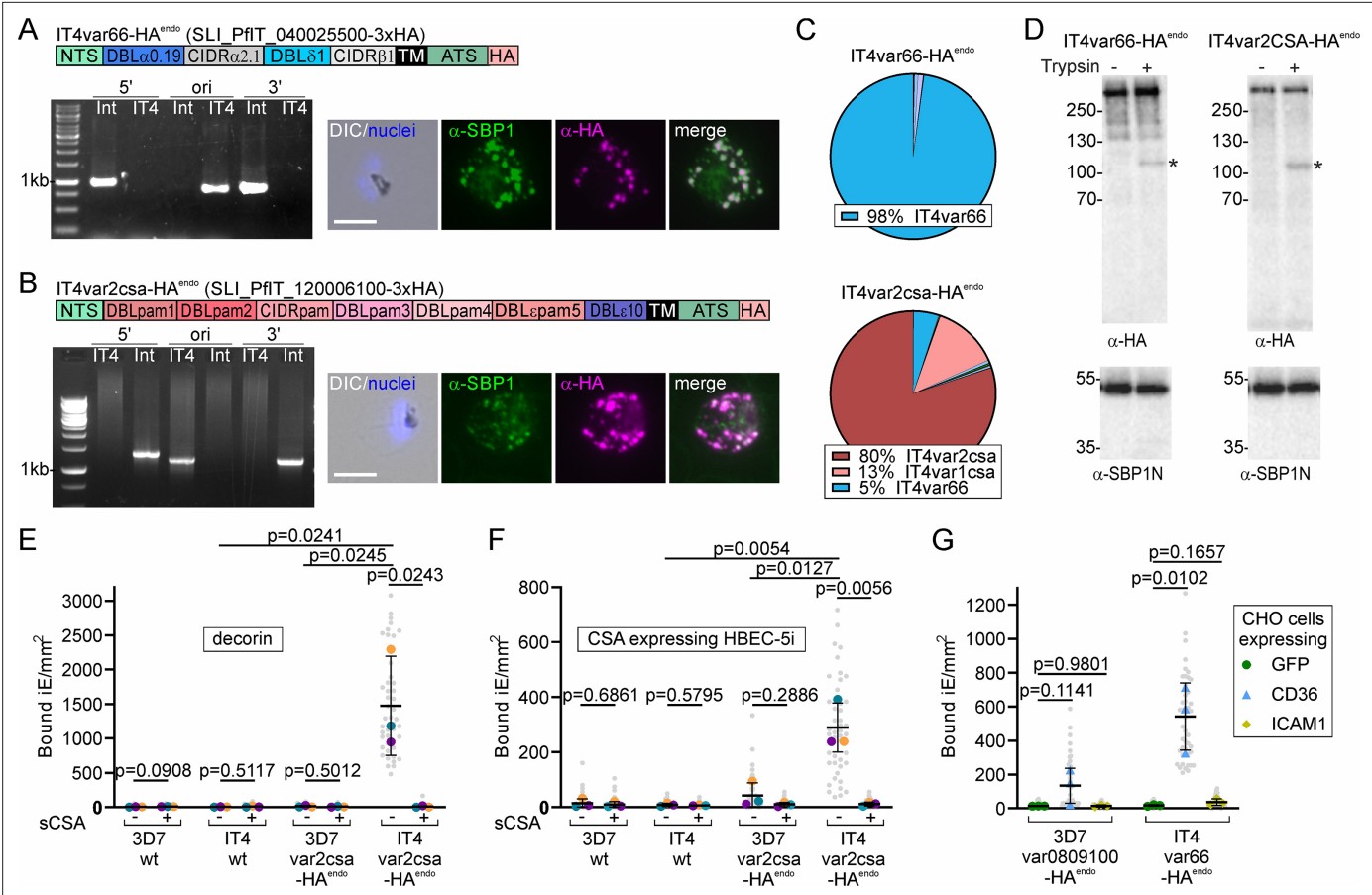

**Figure 3.** Activated PfEMP1s in IT4 are functional and cytoadherent. (**A, B**) Activation of indicated PfEMP1. Scheme shows domain organization. Agarose gel shows PCR products confirming correct integration of the SLI plasmid as described in *Figure 1A*; see *Table 1* for sequence of primers used: IT4: parent; Int: integrant cell line. Fluorescence microscopy images show IFAs with indicated antibodies. Nuclei: Hoechst 33342; DIC: differential interference contrast; size bars 5 μm. (**C**) Pie charts show proportions of total *var* gene transcripts of the indicated cell lines determined by RNAseq (normalized to TPM). (**D**) Western blot of trypsin cleavage assays with indicated parasites. Asterisks show protected PfEMP1 fragment. α-SBP1-N: control for integrity of host. Marker in kDa. (**E, F**) SuperPlots showing binding assays of indicated cell lines against decorin or CSA-expressing HBEC-5i cells (three biological replicates with 15 fields of view/experiment and condition; bars: mean of averages of replicates with SD; unpaired t-test; p-values are indicated). Small gray dots: bound iE/field of view, extrapolated to mm². Larger colored dots: average of bound iE/mm²/replicate. Same color indicates experiment conducted in parallel. iE: infected erythrocytes. (**G**) SuperPlot of binding assays of indicated cell lines against CHO cells expressing GFP, CD36, or ICAM-1 (three biological replicates with 15 fields of view/ experiment and condition; bars: mean of averages of replicates with SD; unpaired t-test; p-values are indicated). Small gray dots: bound iE/field of view, extrapolated to mm². Larger colored dots: average bound iE/mm²/replicate. iE: infected erythrocytes.

The online version of this article includes the following source data and figure supplement(s) for figure 3:

**Source data 1.** Unedited agarose gels are shown in panels A and B.

**Source data 2.** Unedited agarose gels are shown in panels A and B with annotation.

**Source data 3.** Full and unedited blots corresponding to panel D.

**Source data 4.** Full and unedited blots annotated and indicating the regions shown in panel D.

**Source data 5.** Binding assays corresponding to panel E.

**Source data 6.** Binding assays corresponding to panel F.

**Source data 7.** Binding assays corresponding to panel G.

**Figure supplement 1.** Automated scoring pipeline with validation.

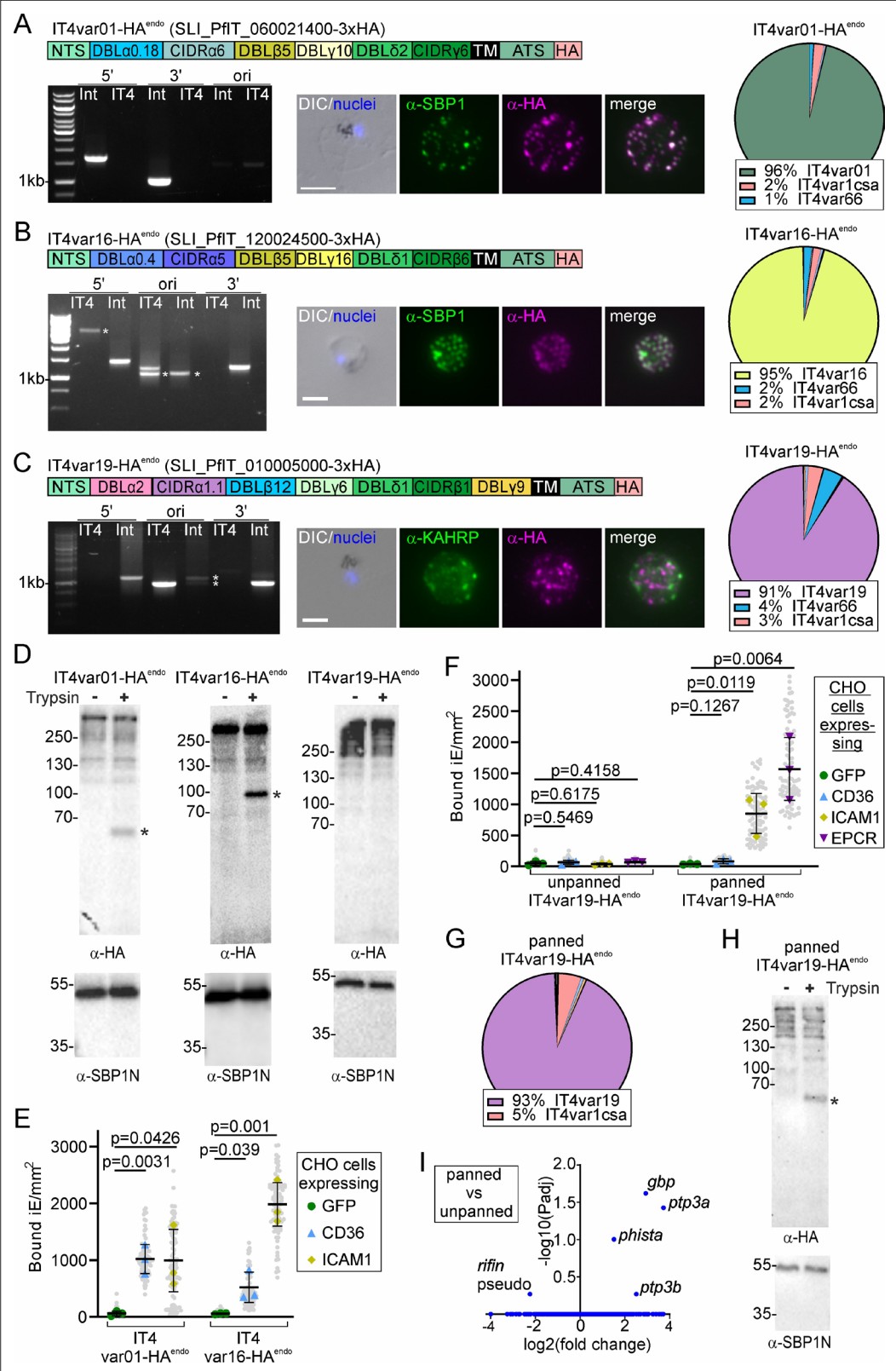

**Figure 4.** Activation of further PfEMP1s with different binding properties in IT4. (**A, B, C**) Activation of indicated PfEMP1. Scheme shows domain organization. Agarose gel shows PCR products confirming correct integration of the SLI plasmid as described in **Figure 1A**; see **Table 1** for sequence of primers used: IT4: parent; Int: integrant cell line. Asterisks indicate non-specific bands; for the original locus, this likely includes bands from other *var*

*Figure 4 continued on next page*

*Figure 4 continued*

genes that result in PCR products of slightly different size to that of the correct *var* gene. Fluorescence microscopy images show IFAs with indicated antibodies. Nuclei: Hoechst 33342; DIC: differential interference contrast; size bars 5 µm. Pie charts show proportions of total *var* gene transcripts of the indicated cell lines determined by RNAseq (normalized to TPM). (**D**) Western blot of trypsin cleavage assays with indicated parasites. Asterisks show protected PfEMP1 fragment. α-SBP1-N: control for integrity of host cell. Marker in kDa. (**E, F**) SuperPlots of binding assays of indicated cell lines against CHO cells expressing GFP, CD36, ICAM-1, or EPCR (3 biological replicates with 15 fields of view/experiment and condition; bars: mean of averages of replicates with SD; unpaired t-test; p-values are indicated). Small gray dots: bound iE/field of view, extrapolated to mm$^2$. Larger colored dots: average of bound iE/mm$^2$/replicate. iE: infected erythrocytes. (**G, H**) Pie chart showing proportions of total *var* gene transcripts as determined by RNAseq (normalized to TPM) and Western blot of trypsin cleavage assay as described in (**D**) of IT4var19-HA$^{endo}$ parasites after five rounds of panning on EPCR. (**I**) Volcano plot showing differential expression analysis (DeSeq2) of EPCR-panned against unpanned IT4var19-HA$^{endo}$ parasites (see ***Source data 1*** for full RNASeq data).

The online version of this article includes the following source data for figure 4:

**Source data 1.** Unedited agarose gels are shown in panels A–C.

**Source data 2.** Unedited agarose gels are shown in panels A–C with annotation.

**Source data 3.** Full and unedited blots corresponding to panel D.

**Source data 4.** Full and unedited blots annotated and indicating the regions shown in panel D.

**Source data 5.** Binding assays corresponding to panel E.

**Source data 6.** Binding assays corresponding to panel F.

**Source data 7.** Full and unedited blots corresponding to panel H.

**Source data 8.** Full and unedited blots annotated and indicating the regions shown in panel H.

parasites (***Figure 4E***). In contrast, the IT4var19-HA$^{endo}$ parasites showed no significant binding to EPCR, CD36, or ICAM-1 (***Figure 4F***). However, after five rounds of panning against EPCR-expressing CHO cells, the panned IT4var19-HA$^{endo}$ parasites exhibited significant binding to EPCR and to a lower degree to ICAM-1 (***Figure 4F***) even though the *var* transcript profile was not noticeably altered compared to the unpanned IT4var19-HA$^{endo}$ parasites and still showed predominant expression of *IT4var19* with similar overall *var* transcript levels (***Figure 4G***; ***Figure 1—figure supplement 1K***). Trypsin assay of the panned parasites showed surface expression of IT4VAR19-HA (***Figure 4H***), contrasting with the unpanned parasites (***Figure 4D***). RNA-Seq indicated only a few genes differently transcribed in the panned compared to the unpanned IT4var19-HA$^{endo}$ parasites (***Figure 4I***). Interestingly, this included the two paralogs of *ptp3* that both were upregulated after panning. The single *ptp3* gene in 3D7 encodes a protein needed for PfEMP1 surface display in 3D7 (***Maier et al., 2008***), suggesting low *ptp3* expression as the reason for failure of the unpanned IT4var19-HA$^{endo}$ parasites to bind. As the two *ptp3* loci in IT4 are more than 10 genes apart, including 3 genes with comparable expression levels (to the *ptp3* genes) that were not differentially expressed in the panned parasites (PfIT_140083600, PfIT_140084100, PfIT_140084200), the initially low expression likely was not due to a genomic deletion but plastically altered transcription of *ptp3* genes.

In summary, we obtained parasites binding to the most common receptors although in the case of EPCR, we had to pan the parasites and detect binding to ICAM-1 in addition to EPCR, which was unexpected (***Avril et al., 2012***; ***Nunes-Silva et al., 2015***; ***Adams et al., 2021***).

## Additional genomic modification in SLI cell lines by using SLI2

To further exploit the SLI-generated PfEMP1 expressor parasites, we generated a second SLI plasmid system with different selection markers termed SLI2 to modify the genome of parasites already carrying a SLI modification (***Figure 5A***). To test SLI2, we used the IT4var01-HA$^{endo}$ line and applied SLI2 to disrupt PTP1 (***Maier et al., 2008***), a Maurer's clefts located PfEMP1 trafficking protein. Integration of the SLI2 plasmid into the correct genomic locus and perpetuation of the first genomic modification was confirmed by diagnostic PCR (***Figure 5B***). Live cell imaging showed a GFP signal in the food vacuole and faint dispersed signal in the host cell, confirming successful inactivation of PTP1 (***Figure 5C***). IFAs showed that in the PTP1-TGD parasites, SBP1 and PfEMP1 were found in many small foci in the host cell that exceeded the average number of ~15 Maurer's clefts typically found

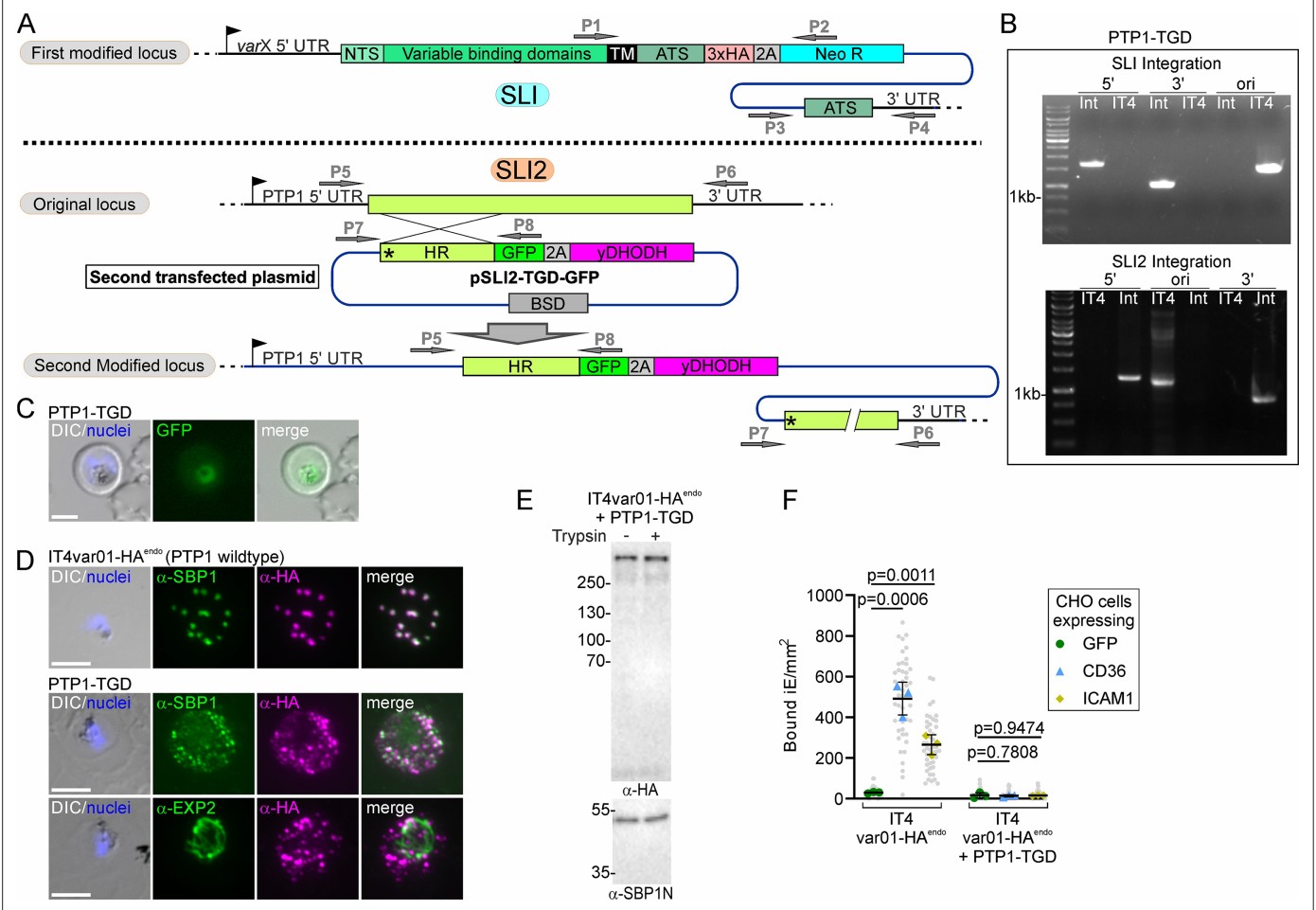

**Figure 5.** Second endogenous modification with SLI2 in a SLI *var* gene cell line. (**A**) Schematic for SLI2 strategy for second genome modification in SLI cell line with activated *var* gene. HR: homology region; ATS: acidic terminal segment; NTS: NTS domain; 2 A: T2A skip peptide; NEO-R: neomycin-resistance gene; yDHODH: yeast dihydroorotate dehydrogenase; BSD: Blasticidin-S-deaminase gene, arrows P1-8 primers for diagnostic PCR; X: desired *var* gene; PTP1: PfEMP1 transport protein 1. (**B**) Agarose gel shows PCR products confirming correct integration of the SLI2 plasmid and perpetuation of the SLI plasmid integration. SLI2 integration: product over 5' integration junction (5'): P5+P8; over 3' integration junction (3'): P7+P6; original locus (ori): P5+P6; SLI integration PCRs as described in *Figure 1A*; IT4: parent; Int: integrant cell line; primers in *Table 1*. (**C**) Fluorescence microscopy images of live IT4var01-HA^endo+PTP1TGD-GFP parasites. (**D**) Fluorescence microscopy images of IFAs with indicated antibodies. Nuclei: Hoechst 33342; DIC: differential interference contrast; size bars 5 μm. (**E**) Western blot of trypsin cleavage assays with IT4var01-HA^endo+PTP1 TGD parasites. α-SBP1-N: control for integrity of host cell. Marker in kDa. (**F**) SuperPlot of binding assays of indicated cell lines against CHO cells expressing GFP, CD36, or ICAM-1 (3 biological replicates with 15 fields of view/experiment and condition; bars: mean of averages of replicates with SD; unpaired t-test; p-values are indicated). Small gray dots: bound iE/field of view, extrapolated to mm². Larger colored dots: average of bound iE/mm²/replicate. iE: infected erythrocytes.

The online version of this article includes the following source data for figure 5:

**Source data 1.** Unedited agarose gels are shown in panel B.

**Source data 2.** Unedited agarose gels are shown in panel B with annotations.

**Source data 3.** Full and unedited blots corresponding to panel D.

**Source data 4.** Full and unedited blots annotated and indicating the regions shown in panel D.

**Source data 5.** Binding assays corresponding to panel F.

per infected RBC (*Blancke Soares et al., 2025*; *Figure 5D*). This phenotype resembled the previously reported Maurer's clefts phenotype of the PTP1 knockout in CS2 parasites (*Rug et al., 2014*). PfEMP1 was still transported into the host cell in the PTP1 disruption parasites (*Figure 5D*) but PfEMP1 was no longer surface exposed (*Figure 5E*) and the parasites failed to bind to CD36 and ICAM-1 (*Figure 5F*), indicating that the IT4var01-HA^endo parasites with the PTP1-TGD had lost their ability for cytoadhesion.

While we did not detect the failure of PfEMP1 transport into the host cell, the binding phenotype agrees with previous work (*Rug et al., 2014*; *Maier et al., 2008*). Hence, SLI2 permits the study of other proteins in SLI generated PfEMP1 expressor lines, for instance, to study trafficking and binding of PfEMP1.

## Proxiome of activated PfEMP1 using BioID

Next, we assessed whether SLI could be used to obtain proxiomes (proximal proteins and interactors) of PfEMP1 in living parasites by generating parasites expressing a PfEMP1 fused with the promiscuous biotin ligase BirA* to carry out BioID (*Roux et al., 2012*). For this, we chose IT4var01 and generated three SLI cell lines with BirA* in different positions of that PfEMP1 (IT4var01-BirA*Pos1[endo], −2[endo] and −3[endo]) (*Figure 6A-D*, *Figure 6—figure supplement 1*). In position 1, BirA* was C-terminal to the ATS, in position 2 between transmembrane domain and ATS, and in position 3 directly upstream of the transmembrane domain (*Figure 6A–D*; note that *IT4var01-BirA*Pos3* lacked the intron (Data S1)). The resulting cell lines predominantly expressed the modified PfEMP1 (as judged by IFA and RNA-Seq), the PfEMP1 was on the surface and the infected RBCs showed the expected binding pattern (*Figure 6B–F*). Parasites expressing the position 3 PfEMP1 construct showed less binding, suggesting partial impairment of placing BirA* into the extracellular part of PfEMP1. Nonetheless, overall the BirA* modified PfEMP1 was functional (*Figure 6F*). BirA* in the PfEMP1 was active as judged by strepta-vidin blots which showed biotinylation with all three cell lines but not with the IT4 parent (*Figure 6G*). Next, we carried out BioID experiments with these cell lines, analyzing enrichment of biotinylated proteins over IT4 in two sequential protein extracts: first the proteins extractable by mild detergents (*Figure 6—figure supplement 1*, *Figure 6—source data 6*) and the proteins requiring extraction with SDS to release more structurally connected proteins (*Figure 6H-J*, *Figure 6—figure supplement 1*, *Figure 6—source data 6*). In all experiments, the tagged PfEMP1 (but no other PfEMP1) was highly enriched due to self-biotinylation. Comparably few proteins were enriched in the mild detergent fraction (*Figure 6—figure supplement 1*), but the SDS-fraction contained many proteins known to be important for PfEMP1 trafficking, including for instance SBP1, MAHRP1, REX1, and several PTPs (1, 2, 4, 5, 7), indicating efficient detection of PfEMP1 trafficking factors (*Figure 6H-J*, *Figure 6—figure supplement 1*). In addition, other exported proteins were detected. This, for instance, included proteins of the MSRP6 complex found at the Maurer's clefts, which is involved in anchoring the clefts but has no role in PfEMP1 transport (*Blancke Soares et al., 2025*), several PHISTs (*Sargeant et al., 2006*), and exported proteins with unknown function here termed EMP1 interacting candidate 1–6 (EMPIC1-6; *Figure 6H-J*, *Figure 6—source data 6*).

Interestingly, the repertoire and relative enrichment of the proteins detected in the BioIDs with the three constructs was remarkably similar (*Figure 6H-J*, *Figure 6—figure supplement 2A*, *Figure 6—source data 6*), including position 3 where BirA* is located on the C-terminal side of the transmem-brane domain. This supports the hypothesis that PfEMP1 is not transported as an integral membrane protein (*Batinovic et al., 2017*; *Marti and Spielmann, 2013*; *Papakrivos et al., 2005*; *Petersen et al., 2016*) as BirA* in the N-terminal part appears to have had access to biotinylate the same proteins as BirA* in the C-terminal part. While there was little evidence for topology-specific interac-tors, several of the detected PHISTs (PfIT_120058000, PfIT_040006400; PfIT_130076100) as well as GEXP10 and the less efficiently enriched PTEF are known to be RBC membrane localized (*Birnbaum et al., 2017*; *Chan et al., 2017*; *Dantzler et al., 2019*; *Davies et al., 2023*; *Hermand et al., 2016*; *Tarr et al., 2014*), indicating that the proxiome also included hits from surface exposed PfEMP1. The hits obtained with the 3 PfEMP1 BirA*-fusion constructs were also more similar to each other than to a general Maurer's clefts proxiome or that of the MSRP6 Maurer's clefts binding domain (*Blancke Soares et al., 2025*), suggesting specificity for PfEMP1 (*Figure 6—figure supplement 2B*). Further, hits from other structures than Maurer's clefts, such as the tether protein MAHRP2 (*Pachlatko et al., 2010*) and the known (*Külzer et al., 2012*; *Zhang et al., 2017*) and likely (*Schulze et al., 2015*) J-dot proteins HSP70x, GEXP18, and PHIST P2 (PfIT_120006500; *Figure 6—source data 6*) further supported that the PfEMP1 proxiome covered hits beyond its dominant location at the Maurer's clefts.

One notable difference between the hits of the 3 PfEMP1 BirA*-fusion constructs was that the position 1 construct detected 7 PHISTs, whereas the position 2 and 3 constructs only detected 3 and 2 PHISTs, of which one was not detected with position 1 (*Figure 6H-J*, *Figure 6—figure supple-ment 2A*, *Figure 6—source data 6*). In addition, some of the PTPs (PTP1 and PTP7) appeared to be

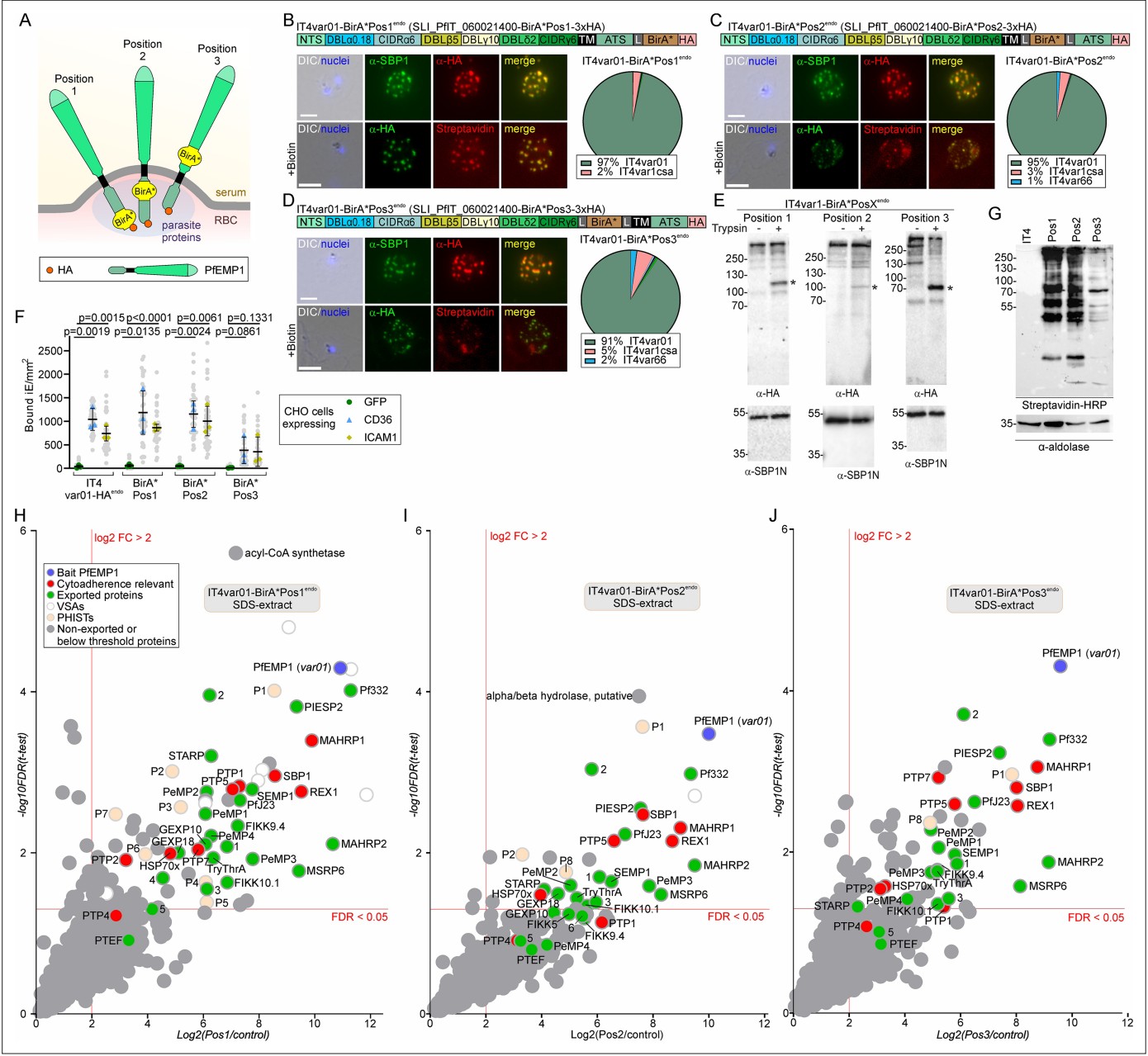

**Figure 6.** Proxiome of PfEMP1 from living parasites. (**A**) Schematic of the three different 3xHA-tagged BirA*-IT4var01 fusion constructs and the position respective to the membrane in the fusion constructs reaching the host cell surface. (**B, C, D**) Confirmation of the activation and modification of the indicated IT4var01-BirA* fusions. Fluorescence microscopy images show IFAs with indicated antibodies or streptavidin. Nuclei: Hoechst 33342; DIC: differential interference contrast; size bars 5 µm. Pie charts show proportions of total *var* gene transcripts of the indicated cell lines determined by RNAseq (normalized to TPM). (**E**) Western blot of trypsin cleavage assays with indicated parasites. Asterisks show protected PfEMP1 fragment. α-SBP1-N: control for integrity of host cell. Marker in kDa. (**F**) SuperPlot of binding assays of indicated cell lines against CHO cells expressing GFP, CD36, or ICAM-1 (3 biological replicates with 15 fields of view/experiment and condition; bars: mean of averages of replicates with SD; unpaired t-test; p-values are indicated). Small gray dots: bound iE/field of view, extrapolated to mm². Larger colored dots: average of bound iE/mm²/replicate. iE: infected erythrocytes. (**G**) Western blot of extracts of the indicated cell lines after incubation with biotin for 24 hr. Streptavidin probes biotinylated proteins; α-aldolase is the loading control. (**H, I, J**) Volcano plots showing enrichment of biotinylated proteins extracted with SDS from the indicated cell lines compared to IT4 wild-type parasites (24 hr growth with biotin; full data in *Figure 6—source data 6*). Only the quadrant with positive enrichment is shown, full plots in *Figure 6—figure supplement 1* and further comparisons in *Figure 6—figure supplement 2*. Hits are color-coded as indicated and short names are given in the plot for known proteins. Phists and other exported proteins without short names were numbered (accessions are found under abbreviations in *Figure 6—source data 6*).

The online version of this article includes the following source data and figure supplement(s) for figure 6:

*Figure 6 continued*

**Source data 1.** Full and unedited blots corresponding to panel E.

**Source data 2.** Full and unedited blots annotated and indicating the regions shown in panel E.

**Source data 3.** Full and unedited blots corresponding to panel G.

**Source data 4.** Full and unedited blots annotated and indicating the regions shown in panel G.

**Source data 5.** Binding assays corresponding to panel F.

**Source data 6.** Mass spectrometry data corresponding to panels H-J.

**Figure supplement 1.** Correct integration of BioID cell lines and full volcano plots.

**Figure supplement 1—source data 1.** Unedited agarose gels are shown in panels A–C.

**Figure supplement 1—source data 2.** Unedited agarose gels are shown in panels A–C with annotations.

**Figure supplement 2.** Volcano plots with color coding for comparison.

differentially enriched in the 3 positions. Finally, position 1 detected the largest number of enriched proteins, possibly because the larger distance from the transmembrane domain permitted more efficient labeling or because this part of the construct is in proximity to a larger number of proteins.

Taken together, these experiments detected most of the proteins previously implicated with PfEMP1 transport and surface display, indicating these proxiomes give a valid representation of proteins in contact with PfEMP1.

## Identification of novel proteins needed for PfEMP1-mediated cytoadhesion

We selected several proteins from the PfEMP1 proxiomes (*Figure 6H-J*, *Figure 6—source data 6*) that previously had not been connected with PfEMP1 transport and used SLI2 to generate full-length tagged versions (*Figure 7—figure supplement 1A*) as well as disruptions (*Figure 7—figure supplement 1B*) by targeting the corresponding genes in the IT4var01-BirA*Pos1endo parasites to assess whether they could be needed for PfEMP1-mediated cytoadhesion. This included PfIT_020007200 that we had previously identified as a PEXEL negative exported protein (PNEP) exported to the host RBC periphery (*Birnbaum et al., 2017*) and that was implicated in VAR2CSA translation and named *P. falciparum* translation enhancing factor (PTEF) (*Chan et al., 2017*). We also included TryThrA (PfIT_080035200), a PNEP (*Heiber et al., 2013*) that was a prominent hit in the BioIDs with all positions, including in the Triton fraction as well as EMPIC3 (PfIT_070007400), also a PNEP (*Heiber et al., 2013*) detected with all 3 PfEMP1 BioID constructs with intermediate to high enrichment (*Figure 6—source data 6*; *Figure 7A*). In addition, we included PeMP2 (PfIT_050006400), a member of the MSRP6 complex (*Blancke Soares et al., 2025*), not previously tested for its function in PfEMP1-mediated cytoadhesion.

Full-length endogenously Ty1-tagged TryThrA, EMPIC3, and PeMP2 showed IFA patterns consistent with the reported localizations of these proteins in the host cell (*Figure 7—figure supplement 1A*; *Birnbaum et al., 2017*; *Blancke Soares et al., 2025*; *Heiber et al., 2013*). For TryThrA-Tyendo and EMPIC3-Tyendo, we observed cells where the Ty1 signal appeared as circles that likely corresponded to the Maurer's clefts periphery and only partially overlapped with the HA signal of PfEMP1 (*Figure 7—figure supplement 1A*). This staining pattern is reminiscent of the subcompartmentalization of different proteins at the Maurer's clefts previously observed by superresolution microscopy (*McMillan et al., 2013*).

Next, we examined the parasites wherein the selected candidates had been disrupted. While the SLI2-based disruptions of PTEF and PeMP2 did not result in loss of parasite binding to CD36 and ICAM1, the TryThrA- and EMPIC3-TGDs resulted in markedly reduced binding (*Figure 7B*), indicating TryThrA and EMPIC3 are novel proteins needed for cytoadhesion. Interestingly, the TryThrA-TGD parasites showed atypical localization of PfEMP1, SBP1, REX1 (dispersed signal in the host cell in addition to foci; disproportionally strong foci) but not KAHRP, pointing to a defect of the Maurer's clefts or protein transport to these structures, while in the EMPIC3-TGD parasites PfEMP1, SBP1, and REX1 showed a localization typical for Maurer's clefts with clearly defined foci and absence of a dispersed pool (*Figure 7—figure supplement 1B*).

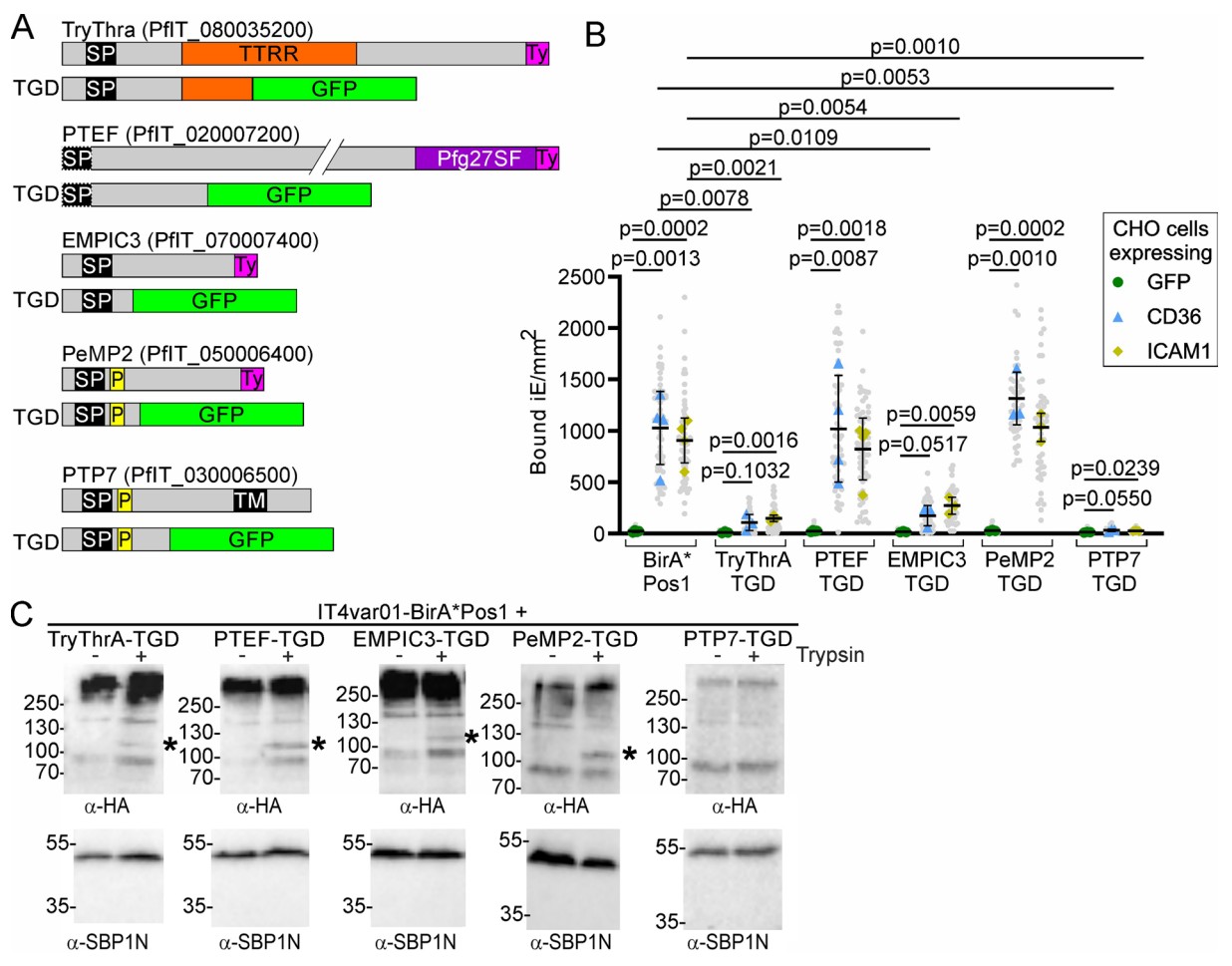

**Figure 7.** New proteins needed for PfEMP1 cytoadherence function. (**A**) Domain schematic of candidates selected for analysis (Ty1-tagging and disruption (TGD) using SLI2) in IT4var01-BirA*Pos1endo. (**B**) SuperPlot of binding assays of the indicated cell lines against CHO cells expressing GFP, CD36, or ICAM-1 (3 or 4 (control and PTEF-TGD) biological replicates with 15 fields of view/experiment and condition; bars: mean of averages of replicates with SD; unpaired t-test; p-values are indicated). Small gray dots: bound iE/field of view, extrapolated to mm$^2$. Larger colored dots: average of bound iE/mm$^2$/replicate. iE: infected erythrocytes. (**C**) Western blot of trypsin cleavage assays with indicated parasites. Asterisks show protected PfEMP1 fragment. α-SBP1-N: control for integrity of host cell. Marker in kDa.

The online version of this article includes the following source data and figure supplement(s) for figure 7:

**Source data 1.** Binding assays corresponding to panel B.

**Source data 2.** Full and unedited blots corresponding to panel C.

**Source data 3.** Full and unedited blots annotated and indicating the regions shown in panel C.

**Figure supplement 1.** Cell lines and analysis of cytoadherence protein candidates.

**Figure supplement 1—source data 1.** Unedited agarose gels shown in panels A and B.

**Figure supplement 1—source data 2.** Unedited agarose gels shown in panels A and B with annotations.

**Figure supplement 2.** Live cell imaging of episomally expressed SBP1-mCherry in the EMPIC3- and TryThrA-TGD parasites.

**Figure supplement 2—source data 1.** Quantification of phenotypes of the EMPIC3- and TryThrA-TGDs corresponding to panels B and C.

In order to better define the phenotype in the TryThrA-TGD parasites, we transfected this and the EMPIC3-TGD parasite line with an episomal plasmid mediating expression of mCherry tagged SBP1, permitting analysis of live, unfixed parasites. As we had already used four selection markers to generate these parasites, we employed a plasmid encoding a mutated version of the lactate transporter FNT (*Wu et al., 2015*, *Marchetti et al., 2015*, *Golldack et al., 2017*) that confers resistance to the chemical BH267.meta (*Walloch et al., 2020*), to transfect these parasites and episomally express SBP1-mCherry (*Figure 7—figure supplement 2A*). While in the EMPIC3-TGD parasites, SBP1-mCherry

was found in foci in the host cell typical for Maurer's clefts (*Figure 7—figure supplement 2B*), the majority of the TryThrA-TGD parasites showed an additional soluble pool of SBP1-mCherry in the host cell (*Figure 7—figure supplement 2C*). Some TryThrA-TGD parasites also contained foci of increased intensity, suggesting enlarged or aggregated Maurer's clefts (*Figure 7—figure supplement 2C*). As the IFAs showed a similar phenotype for REX1 and PfEMP1, the TryThrA-TGD parasites appear to have a defect in the localization of multiple Maurer's clefts proteins and possibly the morphology of the Maurer's clefts. These changes likely are responsible for the cytoadhesion defect in the TryThrA-TGD. In contrast, no obvious changes explaining the binding phenotype were detected in the EMPIC3-TGD parasites.

To ensure the cytoadhesion defect was not due to unrelated changes that had occurred during generation of the TGDs, we sequenced the genome of the TryThrA and EMPIC3 TGD lines. No major changes compared to the cytoadherent IT4var01-BirA*Pos1^endo parent were detected that would explain the cytoadhesion defect (*Source data 2*). Surface trypsin treatment assays showed that in both cases, some PfEMP1 was still surface exposed (*Figure 7C*), indicating that transport to the surface was either merely reduced or that an impairment of the correct presentation of PfEMP1 on the surface caused the binding defect. To ensure this was not due to limitations of the trypsin assay, we also disrupted PTP7, which also was a prominent hit in our BioIDs (*Figure 6I*) and is a well-characterized PfEMP1 trafficking protein that results in loss of surface transport when disrupted (*Carmo et al., 2022*). We confirmed the cytoadhesion defect in the parasite with a disrupted PTP7 (*Figure 7B*), and in contrast to the TryThrA and EMPIC3 TGD cell lines, trypsin assays indicated that there is no IT4VAR01-HA on the surface in that cell line (*Figure 7C*).

In conclusion, we identified two novel proteins (TryThrA and EMPIC3) needed for PfEMP1 function. Given that many of the known PfEMP1 trafficking proteins were detected in the proxiomes and testing some of the others revealed more such proteins, we assume that the BioID experiments give a relevant representation of the protein environment of PfEMP1 and likely contain further proteins important for cytoadhesion.

## Discussion

PfEMP1 is central to the virulence of *P. falciparum* parasites (*Miller et al., 2002*) and the main target of antibody-mediated immunity in symptomatic malaria patients (*Chan et al., 2012*), but studying these important proteins is challenging. Using SLI, we here generated cell lines predominantly expressing a PfEMP1 of choice and show that this facilitates the study of diverse aspects of PfEMP1 biology, including mutually exclusive expression, trafficking, interactome, and receptor binding. A small epitope tag permits reliable tracking of the SLI-targeted PfEMP1, avoiding issues detecting specific variants or the ATS. In addition, we show that larger tags such as a mDHFR domain or BirA* can be added and used to study transport or obtain the proxiome of functional PfEMP1 from living parasites. This also highlights positions in the PfEMP1 sequence where larger tags are tolerated, including in the external region, although the latter reduced the binding efficiency to some extent. Importantly, SLI ensures expression of the modified locus, which would be difficult with other approaches. We further introduce a second SLI system (SLI2) which permits a convenient further genomic modification while maintaining expression of the desired PfEMP1. This will also be of general usefulness to obtain double genome edited parasites.

The generated lines were capable of switching when G418 was lifted, indicating the system can be used to study switching and mutually exclusive expression of *var* genes. However, it should be noted that it is not known whether all mechanisms controlling mutually exclusive expression and switching remain intact in parasites with SLI-activated *var* genes.

Previous work indicated co-activation of genes in a head-to-tail position to the SLI-activated variant gene (*Omelianczyk et al., 2020*). We here only found evidence of co-activation with the activated *var* with genes in a head-to-head orientation, suggesting this occurred due to a shared promoter, rather than a general relaxation of silenced chromatin around the active *var* gene. Similar head-to-head activation had been detected when parasites expressing specific *var* genes were enriched by panning (*Claessens et al., 2012*). However, it is unclear if this can be generalized, and it is possible that different *var* loci respond differently. We also confirmed reduced mutually exclusive expression in a previously published 3D7 cell line (*Joergensen et al., 2010*) that we here termed 3D7^MEED and may be useful to study *var* silencing mechanisms.

PfEMP1-receptor binding and neutralizing antibody mechanisms are increasingly being understood on a structural level and are relevant to understand malaria pathology and effectivity of the immune response in patients (*Rajan Raghavan et al., 2023*; *Reyes et al., 2024*). The straightforward capacity to generate cytoadherent parasite lines uniformly expressing a single PfEMP1 of interest opens up approaches to study receptor-binding as well as antibody-binding and inhibition using native as well as modified PfEMP1. The latter could be done by inserting point mutations, removing, exchanging, or altering domains, for example, by modifications directly in the original SLI plasmid or using CRISPR in the SLI-activated line.

An unexpected finding of this work was that IT4var19-expressing parasites bound ICAM-1 in addition to EPCR as this is considered a PfEMP1 that only binds EPCR (*Avril et al., 2012*; *Nunes-Silva et al., 2015*; *Adams et al., 2021*), although some studies indicated that it may bind additional receptors (*Gillrie et al., 2015*; *Ortolan et al., 2022*). Interestingly, selection for EPCR-binding was required to achieve avid EPCR binding of the *IT4var19* expressor line. While this binding selection did not change the *var* expression profile and IT4var19 remained the dominantly expressed PfEMP1, we cannot exclude that this resulted in other changes that could have led to ICAM1 binding. Selection for EPCR-binding was accompanied by higher expression of *ptp3* genes previously shown to affect PfEMP1 presentation and cytoadhesion (*Maier et al., 2008*), suggesting this as a reason why these parasites did not initially bind. As our findings indicate, this was not due to a genome deletion, this raises the possibility of an additional layer controlling surface display through expression of PTP3 as an accessory factor by binding selection. Thus, the combination of uniform *var* expression and phenotype selection may enable detection of hitherto unrecognized PfEMP1 receptor phenotypes and phenomena controlling PfEMP1 surface display.

In the course of this work, the binding phenotype of the *IT4var19* expressor line remained stable over many weeks without further panning. However, given that initial panning had been needed for this particular line, it might be advisable for future studies to monitor the binding phenotype if the line is used for experiments requiring extended periods of cultivation.

Previous work has indicated that mutants of the different proteins involved in PfEMP1 trafficking block its transport at different points on the way to the RBC surface, including at or before passing into the RBC (*Cooke et al., 2006*; *Maier et al., 2007*; *Rug et al., 2014*; *Maier et al., 2008*). Considering the results here and work on SBP1-disrupted parasites (*Blancke Soares et al., 2025*), none of these proteins seems to influence PfEMP1 before it reaches the Maurer's clefts. This aligns with the location of these proteins, which suggests that they function in the host cell. This would mean that the effect of PTEX inactivation on PfEMP1 transport (*Beck et al., 2014*; *Elsworth et al., 2014*) is likely direct, as the exported PfEMP1-trafficking proteins (if prevented from reaching the host cell due to the PTEX block) would not influence PfEMP1 before it reached the host cell. Together with the result from the stage-specific block of PTEX in this work, the currently most plausible scenario is that PfEMP1 is transported by PTEX, after which other exported proteins are needed for transport to the surface and correct surface display. Why the mDHFR-fused PfEMP1 was not prevented in transport when WR was added is unclear, but may be due to the long region between the transmembrane domain and mDHFR (*Mesén-Ramírez et al., 2016*) or due to the lack of GFP which might contribute to the effectivity of folding stabilized mDHFR to prevent translocation.

While our data indicates PfEMP1 uses PTEX to reach the host cell, this could be expected to have resulted in the identification of PTEX components in the PfEMP1 proxiomes, which was not the case. However, as BirA* must be unfolded to pass through PTEX, it likely is unable to biotinylate translocon components unless PfEMP1 is stalled during translocation. For this reason, a lack of PTEX components in the PfEMP1 proxiomes does not necessarily exclude passage through PTEX.

The PfEMP1 proxiome presented here comprised many of the known proteins required for PfEMP1-mediated cytoadhesion. There was a considerable overlap with the Maurer's clefts proxiome, where many of these proteins are localized. It, however, also included proteins experimentally confirmed to be located at other sites in the host cell, including the host cell membrane. Hence, despite the small number of PfEMP1 molecules displayed at the host cell surface (*Sanchez et al., 2019*), the proxiomes included hits from that site. A protein notably absent from our PfEMP1 proxiomes was the major knob component KAHRP (*Culvenor et al., 1987*; *Pologe et al., 1987*; *Rug et al., 2006*). While this was surprising in light of the original in vitro binding studies (*Oh et al., 2000*; *Waller et al., 1999*; *Waller et al., 2000*), a newer study was unable to detect an interaction of KAHRP with the ATS but found

interaction with PHIST domains (*Mayer et al., 2012*). These findings match our proxiome data which, particularly with the position 1 construct, detected many PHIST proteins and suggests that PHISTs may be in more direct contact with the ATS than KAHRP. This also agrees with recent BioIDs with KAHRP as a bait that did not efficiently detect PfEMP1 whereas PTP4 as bait did (*Davies et al., 2023*).

We here report two new proteins needed for PfEMP1-mediated cytoadhesion. As we still detected some surface exposure of PfEMP1, the cytoadhesion defect was either due to reduced transport to the surface or due to incorrect surface display of PfEMP1. One of the identified proteins, TryThrA, was in a recent study with 3D7 found to be dispensable for cytoadhesion (*Takano et al., 2019*). It is possible that this discrepancy is due to the different *P. falciparum* strains used. In *P. berghei* IPIS3, which belongs to the same group of tryptophan-threonine-rich domain proteins, was recently found to be important for sequestration in rodent malaria (*Gabelich et al., 2022*). Although mouse-infecting malaria parasites do not possess PfEMP1, they do harbor orthologous machinery needed for sequestration, suggesting that virulence factor transport is evolutionary conserved even if the virulence factor is different (*De Niz et al., 2016*). This raises the possibility that tryptophan-threonine-rich domain proteins belong to the conserved core of this machinery, similar to SBP1 and MAHRP1 (*De Niz et al., 2016*). PTEF, selected because of its location at the host cell membrane (*Birnbaum et al., 2017*) and previously linked to VAR2CSA translation (*Chan et al., 2017*), did not influence cytoadhesion of IT4VAR01.

The SLI system does have limitations for the study of *var* and PfEMP1 biology. For example, if the targeted exon 2 region is too similar to that of other *var* genes, the SLI plasmid might insert into an unwanted *var* gene. This can be solved by providing a codon-changed exon 2 region in the SLI plasmid and shifting the targeting sequence upstream where there is high sequence variation. The feasibility of such an approach was shown here by generating the cell lines to insert BirA* into position 2 and 3 of IT4VAR01. Another limitation is that the discovery of PfEMP1-binding to unknown receptors may be difficult if, as seen with the IT4var19-HA[endo] parasites, panning for receptor binding is required to select for that binding. However, as most PfEMP1 will bind CD36 or EPCR, pre-selection on these receptors may enable studies of putative receptor interactions. Alternatively, assuming PTP3 expression is causal and the only factor why the IT4var19-HA[endo] parasites had to be panned, episomal expression of PTP3 could ameliorate this and possibly be used to generally enhance surface display and binding.

## Materials and methods

**Key resources table**

| Reagent type (species) or resource | Designation | Source or reference | Identifiers | Additional information |
|---|---|---|---|---|
| Cell line (*P. falciparum* 3D7) | 3D7var0809100-HA[endo] | This study | See designation | 3D7 SLI var expressor line |
| Cell line (*P. falciparum* 3D7) | 3D7var1200600-HA[endo] | This study | 3D7var2csa-HA[endo] | 3D7 SLI var expressor line |
| Cell line (*P. falciparum* 3D7) | 3D7var0425800-HA[endo] | This study | See designation | 3D7 SLI var expressor line |
| Cell line (*P. falciparum* IT4) | IT4var040025500-HA[endo] | This study | IT4var66-HA[endo] | IT4 SLI var expressor line |
| Cell line (*P. falciparum* IT4) | IT4var120006100-HA[endo] | This study | IT4var2csa-HA[endo] | IT4 SLI var expressor line |
| Cell line (*P. falciparum* IT4) | IT4var060021400-HA[endo] | This study | IT4var01-HA[endo] | IT4 SLI var expressor line |
| Cell line (*P. falciparum* IT4) | IT4var120024500-HA[endo] | This study | IT4var16-HA[endo] | IT4 SLI var expressor line |

*Continued on next page*

*Continued*

| Reagent type (species) or resource | Designation | Source or reference | Identifiers | Additional information |
|---|---|---|---|---|
| Cell line (*P. falciparum* IT4) | IT4var010005000-HA[endo] | This study | IT4var19-HA[endo] | IT4 SLI var expressor line |
| Cell line (*P. falciparum* IT4) | IT4var060021400-BirA*Pos1-HA[endo] | This study | IT4var01-BirA*Pos1[endo] | IT4 SLI var expressor line for BioID with BirA* in position 1 |
| Cell line (*P. falciparum* IT4) | IT4var060021400-BirA*Pos2-HA[endo] | This study | IT4var01-BirA*Pos2[endo] | IT4 SLI var expressor line for BioID with BirA* in position 2 |
| Cell line (*P. falciparum* IT4) | IT4var060021400-BirA*Pos3-HA[endo] | This study | IT4var01-BirA*Pos3[endo] | IT4 SLI var expressor line for BioID with BirA* in position 3 |
| Cell line (*P. falciparum* IT4) | IT4var060021400-BirA*Pos1-HA[endo] with SLI2 TryThrA-TGD | This study | IT4var01-BirA*Pos1+TryThrA-TGD | IT4 SLI var expressor line with SLI2-mediated disruption of TryThrA |
| Cell line (*P. falciparum* IT4) | IT4var060021400-BirA*Pos1-HA[endo] with SLI2 EMPIC3-TGD | This study | IT4var01-BirA*Pos1+EMPIC3-TGD | IT4 SLI var expressor line with SLI2-mediated disruption of EMPIC3 |
| Cell line (*P. falciparum* IT4) | IT4var060021400-BirA*Pos1-HA[endo] with SLI2 PTP1-TGD | This study | IT4var01-BirA*Pos1+PTP1-TGD | IT4 SLI var expressor line with SLI2-mediated disruption of PTP1 |
| Cell line (*P. falciparum* IT4) | IT4var060021400-BirA*Pos1-HA[endo] with SLI2 PTEF-TGD | This study | IT4var01-BirA*Pos1+PTEF-TGD | IT4 SLI var expressor line with SLI2-mediated disruption of PTEF |
| Cell line (*P. falciparum* IT4) | IT4var060021400-BirA*Pos1-HA[endo] with SLI2 PeMP2-TGD | This study | IT4var01-BirA*Pos1+PeMP2-TGD | IT4 SLI var expressor line with SLI2-mediated disruption of PeMP2 |
| Cell line (*P. falciparum* IT4) | IT4var060021400-BirA*Pos1-HA[endo] with SLI2 PTP7-TGD | This study | IT4var01-BirA*Pos1+PTP7-TGD | IT4 SLI var expressor line with SLI2-mediated disruption of PTP7 |
| Antibody | Monoclonal rat anti-HA (clone 3F10) | Roche | 11867423001, RRID:AB_390918 | IFA (1:2000), WB (1:1000) |
| Antibody | Monoclonal rabbit anti-HA (C29F4) | Cell Signalling Technologies | 3724, RRID:AB_1549585 | IFA (1:1000) |
| Antibody | Monoclonal mouse anti-Ty1 (clone BB2) | Thermo | MA5-23513, RRID:AB_2610644 | IFA (1:20,000) |
| Antibody | Monoclonal rat anti-RFP (5F8) | Chromotek | 5f8 – 100, RRID:AB_2336064 | IFA (1:1000) |
| Antibody | Monoclonal mouse anti-GFP | Roche | 11814460001, RRID:AB_390913 | IFA (1:1000) |
| Antibody | Polyclonal rabbit anti-GFP | Thermo | A-6455, RRID:AB_221570 | IFA (1:1000) |
| Sequence-based reagent | PCR primer Neo40 (P2) | This paper | P2 | CGAATAGCCTCTCCACCCAAG |
| Sequence-based reagent | PCR primer pARL55 (P3/P7) | This paper | P3/P7 | GGAATTGTGAGCGGATAACAATTTCACACAGG |
| Sequence-based reagent | PCR primer GFP85 (P8) | This paper | P8 | ACCTTCACCCTCTCCACTGAC |
| Sequence-based reagent | PCR primer Ty1 (P8) | This paper | P8 | GTGGATCTTGATTTGTATGC |
| Other | Biotin | Sigma Aldrich | B4639 | BioIDs |
| Other | Hoechst 33342 | Cayman | K9061 | Live cell and IFA DNA stain |
| Other | 4',6-diamidino-2-phenylindole (DAPI) | Roche | 10236276001 | IFA DNA stain |

## Cloning of plasmid constructs

For genome integration constructs, homology regions encoding the C-terminus of target genes (for C-terminal tagging) or a region in the N-terminal part (for TGDs) were PCR amplified from 3D7 or IT4 gDNA purified with the Monarch gDNA Purification Kit, NEB (T3010), or QIAGEN DNA extraction kit (56304) and cloned into pSLI (*Birnbaum et al., 2017*) or pSLI2 using Gibson assembly (*Gibson et al., 2009*) or T4 ligase. Plasmids, including the SLI2 plasmids and the FNT resistance plasmid for episomal expression of SBP1-mCherry, are shown in *Supplementary file 1*. For the Position 1 BirA* fusion plasmids, the targeting region followed by a 7xGGGS linker, a previously used sequence encoding BirA* (*Birnbaum et al., 2020*) and a 3xHA-tag was cloned into pSLI. For position 2 and position 3 plasmids, the part of PfEMP1 encoded C-terminal to BirA* was synthesized with a different codon usage (GenScript) to prevent integration into the genome in that region and was cloned together with the targeting region (Position 2: until amino acid 2415; Position 3: until amino acid 2376), the region encoding BirA* flanked by short linkers and a 3xHA-tag into pSLI. For the episomal early stage blocking construct SBP1-mDHFR-GFP was cloned into pARL2 containing a *mal7* promoter (*Grüring et al., 2011*). For the tagging of PfEMP1 with mDHFR, homology regions encoding the C-terminus of the target *var* genes were cloned into pSLI with a mDHFR domain between the targeting region and a 3xHA-tag. To ensure there were no undesired mutations, all cloned inserts were sequenced by Sanger sequencing (Microsynth).

## Parasite culture

*P. falciparum* parasites (3D7 *Walliker et al., 1987* and IT4 *Jensen and Trager, 1978*) were cultured using standard procedures (*Trager and Jensen, 1976*). The parasites were maintained in RPMI1640 supplemented with 0.5% Albumax (Life Technologies, 11021) and human 0+ erythrocytes (University Medical Center Hamburg-Eppendorf (UKE), Germany) at a hematocrit of 5% at 37 °C under an atmosphere consisting of 1% $O_2$, 5% $CO_2$, and 94% $N_2$.

## Transfection, SLI, and confirmation of correct genome integration

Late schizont stage parasites were purified using Percoll as described (*Rivadeneira et al., 1983*), using 60% Percoll for 3D7 and 64% for IT4 parasites. Fifty µg of purified plasmid DNA (QIAGEN, 12143) were transfected using the Amaxa system (Lonza Nucleofector II AAD-1001N, program U-033) following previously described protocols (*Moon et al., 2013*). Transfectants were selected with either 4 nM WR99210 (Jacobus Pharmaceuticals; pSLI) or 2 µg/ml blasticidin S (Life Technologies, R21001; pSLI2). SLI for selection of parasites with the plasmid integrated into the genome was done as described (*Birnbaum et al., 2017*) by adding 400 µg/ml G418 (Sigma Aldrich, A1720; pSLI) or 0.9 µM DSM1 (Merck, 5.33304.0001; pSLI2) to the culture. After the parasitemia recovered under drug selection, genomic DNA was isolated and correct genomic integration of the plasmid in the knock-in parasites was verified by PCR as described (*Birnbaum et al., 2017*). For transfection of the plasmid harboring the mutated gene encoding PfFNT (amino acid change G 107 S; *Golldack et al., 2017*), BH267.meta (*Walloch et al., 2020*) was used at 5 µM until parasites appeared, after which the concentration of drug was dropped to 2.5 µM to maintain the culture.

## Immunofluorescence and streptavidin-fluorescence assay

IFAs were performed as described (*Spielmann et al., 2003*). Briefly, pelleted parasites (2000 × *g* for 5 min) were washed with 1 x PBS and applied at a hematocrit of 1–2.5% to 10-well glass slides, air-dried, and fixed in acetone for 30 min. Wells were rehydrated with 1 x PBS, then washed five times with 1 x PBS. Antibodies were applied in 1 x PBS containing 3% BSA. Primary antibodies were rat anti-HA (Roche, 11867423001), 1:2000; rabbit anti-HA (Cell Signaling Technology, 3724), 1:1000; rabbit anti-SBP1-C (*Mesén-Ramírez et al., 2016*), 1:2500; rabbit anti-KAHRP (kind gift of Prof. Brian Cooke), 1:500; mouse anti-EXP2 (European Malaria Reagent Repository) used 1:500; rabbit anti-REX1 (*Mesén-Ramírez et al., 2016*), 1:10,000; mouse anti-Ty1 (Thermo, MA5-23513), 1:20,000; rat anti-RFP (Chromotek, 5f8 – 100), 1:1000; mouse anti-GFP (Roche, 11814460001) used 1:1000 and rabbit anti-GFP (Thermo, A-6455), 1:1000. For secondary antibodies anti-rabbit conjugated with Alexa Fluor-488 (A-27934), Alexa-Fluor546 (A-10040), or Alexa Fluor-647 (A-21244), anti-mouse conjugated with Alexa Fluor-488 (A-11001), goat anti-rat conjugated with Alexa Fluor-488 (A-10041), or Alexa Fluor-594 (A-11007) (all Invitrogen) were used (all 1:2000). Secondary antibodies were applied together with

4',6'-diamidine-2'-phenylindole dihydrochloride (DAPI, 10236276001; 1 µg/ml) or Hoechst (50 ng/ml; Cayman, K9061; as indicated in the figure legends) for staining of parasite nuclei. For the Streptavidin-fluorescence assay, streptavidin coupled to Alexa Fluor-594 (Invitrogen, S32356) was added (1:2000) together with the secondary antibody. Slides were mounted with Dako (Sigma Aldrich, S3023) and covered with a cover slip.

## Fluorescence microscopy imaging

Live or fixed parasites were imaged with a Zeiss AxioImager M1 or M2 equipped with a Hamamatsu Orca C4742-95 camera using a 100×/1.4-numerical or a 63×/1.4-numerical aperture lens. AxioVision software (version 4.7) was employed to capture the images. Live cell imaging of parasites expressing fluorescent proteins was performed as previously described (*Grüring and Spielmann, 2012*). To stain the parasites' DNA, parasites were incubated with either 1 µg/ml of DAPI (Roche, 10236276001) or 50 ng/ml Hoechst 33342 (Cayman, K9061) (as indicated in the figure legends) in parasite medium for 10 min at 37 °C. Images were processed in Corel Photo-Paint (version 2021) and arranged in Corel Draw (version 2021).

## Trypsin assay to assess PfEMP1 surface exposure

Parasite cultures with 5–10% parasitemia were synchronized for rings using sorbitol (*Lambros and Vanderberg, 1979*) and then grown for 12 hr at 37 °C. The resulting trophozoite stage parasites were isolated with a Percoll gradient as described (*Heiber and Spielmann, 2014*) for 3D7 cell lines. For IT4 parasites, an adjusted gradient with 80%, 64%, and 40% Percoll was used. The purified infected erythrocytes were washed and split into two samples. One sample was incubated with 50 µg/ml TPCK-treated Trypsin (Sigma Aldrich, 4352157) in 1 x PBS at 37 °C for 30 min while the other sample (control) was incubated in 1 x PBS alone. Thereafter, trypsin inhibitor from soybean (Sigma Aldrich, 10109886001) was added (1 mg/ml final concentration), and the samples were incubated on ice for 15 min. The cells were washed in 1 x PBS, then lysed in 100 µl lysis buffer (4% SDS, 0.5% Triton X-100 in 0.5 x PBS), containing 1 mg/ml trypsin inhibitor, 1 mM PMSF (Thermo Fisher Scientific, 36978), and 1 x complete protease inhibitor cocktail (Roche, 11697498001). Extracts were immediately subjected to SDS-PAGE or frozen at –20 °C until needed.

## Binding assays

For binding assays, Chinese Hamster Ovary (CHO-745 or CHO-K1) cells that express CD36, ICAM-1, GFP [66] or EPCR (in CHO-K1) (*Avril et al., 2016*), or human brain endothelial cells HBEC-5i cells (American Type Culture Collection (ATCC), Manassas, VA, USA; no. CRL-3245) were seeded two (1x10^5 cells/ml) or three (2x10^5 cells/ml) days before the binding assay into a 24-well plate containing coverslips (0.5 ml/well). For binding assays against decorin (chondroitin sulfate proteoglycan from bovine articular cartilage previously used for VAR2CSA binding *Dahlbäck et al., 2011*), the coverslips in 24-well plates were incubated overnight at 4 °C with decorin solution (5 µg/ml in PBS), thereafter washed with 1 x PBS, blocked with 1% BSA in 1 x PBS for 2 hr and washed with 1 x PBS twice (*Renn et al., 2021*). Knobby parasites of the tested cell lines were enriched using 1% gelatin in glucose-free RPMI (16.4 g/l RPMI-HEPES (Applichem, A1538), 0.05 g/l hypoxanthine, 30 ml/l NaHCO$_3$ (7.5 %) and 250 µl/l gentamycin (Ratiopharm, 3928180) in H$_2$O, pH 7.2) as described (*Goodyer et al., 1994*). After washing in binding medium (16.4 g/l RPMI-HEPES and 20 g/l glucose in H$_2$O, pH 7.2), number of erythrocytes/ml (Neubauer counting chamber) and % infected erythrocytes (Giemsa smears) were determined and the suspension adjusted to 2x10^6 infected erythrocytes/ml in binding medium. The wells with the CHO or HBEC-5i cells were incubated with binding medium for 30 min before the parasite suspension was added to the wells (500 µl/well). Per experiment, three wells per parasite cell line and receptor were used. In the binding assays with decorin and HBEC-5i, the parasites' suspension was split and either incubated with soluble CSA (100 µg/ml) or soluble BSA (100 µg/ml) (control) for 30 min at 37 °C before adding the infected erythrocytes to the wells. The plates were then incubated for 60 min at 37 °C for binding, with careful shaking every 15 min. The coverslips were washed six times by carefully dunking them into binding medium and blotting excess medium on paper after every dunk. The coverslips were then laid face-down parallel to the table in a washing plate that was angled at 45° (with the face-side hanging free in the binding medium) and incubated for 30 min at room temperature. Immediately after, the coverslips were fixed in 1% glutaraldehyde in 1 x PBS for 30 min and stained with filtered

10% Giemsa (Merck, 1092040500) in 1 x PBS for 15 min. The stained coverslips were washed in water and glued with CV-Mount (Leica, 14046430011) face-down onto glass slides. Five images per coverslip (per experiment 15 images per parasite line and condition) were captured with a Thermo Fisher EVOS xl (75% light intensity at ×40 magnification).

## Automated counting of binding assays

The evaluation of images of binding assays was automated using Ilastik v1.3.3post3 (*Berg et al., 2019*) and CellProfiler v4.2.1 (*Stirling et al., 2021*; *Figure 3—figure supplement 1*). First, the images of the binding assays were processed with a trained Ilastik model for the segmentation of the foreground (infected erythrocytes) and background (CHO/HBEC-5i cells and plastic). For the training, the pixel classification module was manually trained with 20 microscopy images representing different shapes of infected erythrocytes, backgrounds, and artefacts. All the color/intensity, edge, and texture features were enabled for training. The resulting processed images were exported as probability images with pixel intensities from 0.0 to 1.0 for the probability of a foreground pixel (regression values; 1.0=100% probability for foreground pixel). Ilastik pre-processed images were then fed to a CellProfiler pipeline (*Figure 3—figure supplement 1*) using the 'IdentifyPrimaryObjects' module to identify and count roundish objects with a diameter of 15–35 pixel units. Robust background thresholding and de-clumping by shape was selected. The number of counted infected erythrocytes scored per image was given out as a spreadsheet. To show the reliability of the automated pipeline in comparison to the manual scoring, statistical tests between the two methods were conducted as shown in *Figure 3—figure supplement 1*.

## RNA-Seq and qPCR analysis

Synchronous ring-stage parasites with a parasitemia of 3–5%, from 10 ml of culture were pelleted at 800 × g and dissolved in five pellet volumes of Trizol (Thermo Fisher, 15596018), thoroughly mixed, incubated for 5 min at 37 °C and immediately stored at –80 °C until RNA isolation. To purify the RNA, the Trizol sample was thawed, 1/5 volume of chloroform added, thoroughly mixed, and centrifuged at 16,000 × g for 30 min at 4 °C. The resulting clear supernatant was transferred to a new tube and processed using the Qiagen miRNeasy Mini Kit (217004) according to the manufacturer's instructions. RNA integrity was assessed using the Agilent 2100 bioanalyzer system with the RNA 6000 Pico Kit (Agilent, 5067–1513). All samples had a RIN >8, our cutoff for inclusion.

Ribosomal RNA was removed using QIAseq FastSelect RNA Removal Kit (QIAGEN, 333390). Libraries were prepared with the QIASeq Stranded mRNA Library Kit (QIAGEN, 180440) and sequenced on an Illumina NextSeq 550 system with NextSeq 500/550 Mid Output Kit v2.5 (Illumina, 20024906; 150 cycles). Raw reads were mapped with hisat2 (version 2.2.1) to the respective reference genomes sourced from PlasmoDB (*Amos et al., 2022*; IT4: Release 58; 3D7: Release 62). Mapped reads were sorted and indexed with samtools (version 1.17). Reads mapped to genomic features were counted using featureCounts (version 2.0.4). For *var* genes, only reads mapping to exon 1 were considered; for *rif*s, reads to the entire coding region were included. The data have been deposited in NCBI's Gene Expression Omnibus (*Edgar et al., 2002*) and are accessible through GEO Series accession number GSE267413. Python3 (version 3.11.4) and bioinfokit (version 2.1.2) were used to normalize the reads to transcripts per million (TPM) as well as to create the coverage plots with matplotlib (version 3.7.2). A volcano plot was done in GraphPad Prism. Differential gene expression analysis for panned against unpanned parasites was performed in R with the DESeq2 (version 1.42.0) package.

Quantification of *var* and *rif* transcript levels was measured relative to internal control gene seryl-tRNA synthetase by real-time quantitative PCR using primers specific to each 3D7 *var* or *rif* gene as previously described (*Wang et al., 2009*).

## Assays to analyze PfEMP1 transport into the host cell

Assays assessing the transport of PfEMP1 fused directly to mDHFR (*Eilers and Schatz, 1986*) were done as described (*Grüring et al., 2012*) with some modifications: schizont stages of the corresponding lines were purified with Percoll and allowed to invade for 8 hr, followed by synchronization with 5% sorbitol (*Lambros and Vanderberg, 1979*) to obtain ring stages with an age of 0–8 hr post invasion, the culture split into one with 4 nM WR and one without (control) and grown for 24 hr before analysis by IFA. For co-blocking assays (*Mesén-Ramírez et al., 2016*), where transport through PTEX

was assessed indirectly by conditionally clogging it with another exported protein fused to mDHFR, the parasite cultures were synchronized using Percoll to obtain schizonts as described (*Rivadeneira et al., 1983*) and grown for 24 hr in the presence or absence of 4 nM WR followed by analysis of export by live cell imaging or IFA. For the late stage PTEX block, the pARL2-SBP1-mDHFR-GFP-2A-KAHRP-mScarlet plasmid was utilized (*Mesén-Ramírez et al., 2016*).

## BioID, mass spectrometry, and data analysis

For proximity biotinylation, biotin (Sigma Aldrich, B4639; 50 µM final) was added to asynchronous parasites expressing the BirA*-PfEMP1 fusion constructs as well as to IT4 parent parasites (5% parasitemia, 150 ml per condition and experiment) and cultured for 24 hr with one exchange of medium with fresh biotin after 12 hr. Thereafter, the parasites were washed twice with DPBS before they were subjected to saponin lysis (0.03% saponin in DPBS) on ice for 10 min, followed by five washes in DPBS before lysis in 2 ml lysis buffer (50 mM Tris-HCl pH 7.5, 500 mM NaCl, 1% Triton-X-100, 1 mM DTT, 1 mM PMSF and 1 x protease inhibitor cocktail) and storage at –80 °C. For isolation of proteins, the samples were thawed and frozen two times before centrifugation at 16,000 × *g* for 10 min. The supernatant (Triton-extract) was saved and the pellet frozen, thawed, and once more extracted using 4% SDS in 50 mM Tris-HCl pH 7.5, 500 mM NaCl, 1% Triton-X-100, 1 mM DTT (SDS-extract). The SDS extract was transferred to a fresh tube and cleared by centrifugation at 16,000 × *g* for 10 min. For the purification of biotinylated proteins, both extracts (Triton and SDS) were diluted 2:1 in 50 mM Tris-HCl and incubated with 50 µl Streptavidin Sepharose (GE Healthcare, 17-5113-01) overnight at 4 °C while rotating. The beads were washed twice in lysis buffer, once in H$_2$O, twice in Tris-HCl pH 7.5, and three times in 100 mM Triethylammonium bicarbonate buffer. The proteins on the beads were digested as described (*Hubner et al., 2015*). Briefly, the beads were treated with 50 µl elution buffer (2 M Urea in 100 mM Tris pH 7.5 containing 10 mM DTT) at room temperature, shaking for 20 min. Subsequently, iodoacetamide (IAA) was added to a final concentration of 50 mM and the samples were further incubated in the dark, shaking for 10 min. The proteins were then treated with 0.25 µg Trypsin/LysC (Promega, V5072), while shaking at room temperature. After 2 hr, the supernatants containing eluted proteins were collected and the beads were immersed with an extra 50 µl of elution buffer for 5 min at room temperature. The supernatant was pooled with the previous elution, and the final 100 µl of eluted proteins was supplemented with 0.1 µg of Trypsin/LysC and treated overnight while shaking at room temperature. The protein samples were then desalted on Stagetips using C18 membranes (*Rappsilber et al., 2007*) and eluted in 80% acetonitrile, 0.1% Formic acid.

The acetonitrile was evaporated in a SpeedVac, and the concentrated sample was then reconstituted to a final volume of 12 µl with 0.1% Formic acid. To analyze the sample by mass spectrometry, 5 µl of sample was analyzed during a 60 min run on an Easy-nLC 1000 (Thermo Fisher Scientific) with a 30 cm C18-reverse phase column coupled on-line to an Orbitrap Exploris 480 mass spectrometer (Thermo Fisher Scientific). Data was acquired in top 20 mode with a dynamic exclusion of 45 s.

Raw mass spectrometry data were processed using MaxQuant (*Cox and Mann, 2008*; version 1.6.6.0). Parameters were set to default except for the following: Deamidation (NQ) was added as a variable modification together with oxidation (M) and acetyl (N-term). Match-between-runs and re-quantify options were enabled with default parameters and iBAQ values were calculated. Mass spectra were compared to peptide masses from the *Plasmodium falciparum* IT4 annotated proteome (PlasmoDB v64). The 'proteinGroups' file from MaxQuant output was analyzed using the Perseus software package (*Tyanova et al., 2016*; version 1.4.0.20). The data were filtered against peptides assigned as 'only identified by site', 'reverse' and/or 'potential contaminant' hits in the datasets. IBAQ values were transformed to log$_2$ values and missing values were imputed following a normal distribution. Data obtained from Triton extraction and SDS extraction were analyzed separately. Significant outliers were identified at each position by using the two-sided Benjamini-Hochberg test with an FDR cut-off of 0.05. The mass spectrometry proteomics data have been deposited to the ProteomeXchange Consortium via the PRIDE (*Perez-Riverol et al., 2022*) partner repository with the dataset identifier PXD052297.

## Western blot analysis

Western blots were conducted as described (*Heiber and Spielmann, 2014*). In brief, preparation of extracts from the BioID experiments or the trypsin cleavage assays was centrifuged at 16,000 × *g*

and the supernatant was mixed with 4 x Laemmli sample buffer. Samples were incubated for 10 min at 90 °C before they were applied to 10% polyacrylamide gels for sodium dodecyl sulfate polyacrylamide gel electrophoresis. The proteins separated on the gels were transferred to nitrocellulose membranes (Amershan Protran membranes, GE Healthcare, GE10600002) using transfer buffer (0.192 M Glycine, 0.1% SDS, 25 mM Tris and 20% methanol in $H_2O$). For the detection of proteins by antibodies, membranes were blocked in 5% skim milk in 1 x TBS (50 mM Tris and 150 mM NaCl in $H_2O$) for 2 hr at room temperature, washed three times with 1 x TBS with 1% Tween, and incubated in 1 x TBS with 3% skim milk with the first antibody rolling overnight at 4 °C. First antibodies were rat anti-HA (Roche, 11867423001; 1:1000); rabbit anti-SBP1-N (1:4000; *Mesén-Ramírez et al., 2016*) or rabbit anti-aldolase (1:4000; *Mesén-Ramírez et al., 2016*). Secondary antibodies were horseradish peroxidase (HRP)-conjugated anti-rat (Dianova, 112035003; 1:2000) or HRP-conjugated anti-rabbit (Dianova, GtxRb-003-DHRPX; 1:2000) and were applied in 1 x TBS with 3% skim milk and incubated rolling for 2 hr at room temperature. For the detection of biotinylated proteins, HRP-conjugated streptavidin (Thermo Fisher Scientific, A-11001) was used in 5% BSA in 1 x TBS as described (*Cui and Ma, 2018*) and incubated by rolling overnight at 4 °C. After secondary antibody or HRP-conjugated streptavidin incubation, the membrane was washed three times in 1 x TBS with 1% Tween, then 5 ml ECL solution A (0.025% luminol (Sigma Aldrich, A8511) in 0.1 M Tris-HCl in $H_2O$, pH 8.6) was mixed with 500 μl ECL solution B (6.7 mM p-Coumaric acid in DMSO) and 1.5 μl $H_2O_2$ and applied to the membrane before the ECL signal was detected with a ChemiDoc XRS imaging system (Bio-Rad).

## Whole genome sequencing and analysis

Genome sequencing was done essentially as described (*Behrens et al., 2024*). The NEB Monarch Genomic DNA Purification Kit (T3010) was used to prepare genomic DNA from 50 ml cultures of the TryThrA and EMPIC3 TGD parasites (both generated in the IT4var01-BirA*Pos1[endo] background) and from the parent (IT4var01-BirA*Pos1[endo]). BGI TECH SOLUTIONS (Hong Kong) carried out DNBSEQ PE100 sequencing and bioinformatic analysis. This included calling of SNP, InDel, SV, and CNV compared to IT4 reference. The data was deposited at GEO (Accession number GSE275671) which also includes technical details on sample preparation and filtering. All SNPs leading to a stop or potential splice mistake, all INDELs leading to frame shifts, all SVs and CNVs indicating gene or partial gene loss in the Var01-TGD parasites that were not present in the parent (IT4-Var01 parasites) were manually assessed by inspecting the reads in that region. Only changes affecting exported proteins were considered and were manually re-assessed in all three lines by analyzing the individual reads. In addition, known PfEMP1 trafficking genes were manually checked for differences.

## Quantification, statistical analysis, and figure construction

P values are indicated in the figure and $p<0.05$ was considered as significant. All error bars shown are standard deviations. Statistical significance was determined by unpaired t-test. A ratio-paired t-test was used for the comparison between the individual images of the binding assays evaluated by manual scoring and the automated pipeline. Statistical analysis was done in GraphPad Prism (version 9). Intraclass correlation coefficient (ICC) was calculated using Excel (Microsoft); Two-factor ANOVA without replication was applied; ICC was calculated with the variations of the ANOVA; ICC = $MS_{Row}$-$MS_{Error}$/$MS_{Row}$ +$df_{Column}$x$MS_{Error}$ + ($df_{Column}$ +1)x($MS_{Column}$-$MS_{Error}$)/($df_{Row}$ +1). Graphs were done in GraphPad (version 9) and transferred to CorelDraw (version 2021) with adjustments to style without altering the data. Corel Draw (version 2021) was used to prepare the figures.

## Acknowledgements

We thank Jacobus Pharmaceuticals for WR99210. We thank Eric Beitz for providing BH267.meta. This work was funded by the Joachim Herz Stiftung (Graduate School: Infection biology of tropical pathogens) (JC, TS, IB), the German Research Foundation grant BR 1744/17-1, SP1209/4-1 (IB, TS), the Jürgen Manchot Stiftung, Germany (JS), the Lundbeck Foundation grant R344-2020-934 (TL), the Independent Research Fund Denmark grant 9039–00285 A (TL), Colciencias Scholarship, Colombia (INP), the Leibniz Collaborative Excellence grant K328/2020 (TS, RB), the European Research Council (ERC) under the European Union's Horizon 2020 research and innovation programme (grant agreement No. 101021493) (PLB, TS) and the Deutscher Akademischer Austauschdienst (DAAD) (CCP). The funders had no influence on the project.

# Additional information

## Funding

| Funder | Grant reference number | Author |
| --- | --- | --- |
| Joachim Herz Stiftung | | Jakob Cronshagen<br>Iris Bruchhaus<br>Tobias Spielmann |
| Deutsche Forschungsgemeinschaft | 1744/17-1 | Iris Bruchhaus |
| Deutsche Forschungsgemeinschaft | SP1209/4-1 | Tobias Spielmann |
| Jürgen Manchot Stiftung | | Jan Stäcker |
| Lundbeck Foundation | R344-2020-934 | Thomas Lavstsen |
| Independent Research Fund Denmark | 9039-00285A | Thomas Lavstsen |
| Colciencias Scholarship | | Isabel Naranjo-Prado |
| Leibniz Collaborative Excellence | K328/2020 | Richárd Bártfai<br>Tobias Spielmann |
| European Research Council | grant agreement No. 101021493 | Patricia López-Barona<br>Tobias Spielmann |
| Deutscher Akademischer Austauschdienst | | Carolina Castro-Peña |

The funders had no role in study design, data collection and interpretation, or the decision to submit the work for publication.

## Author contributions

Jakob Cronshagen, Conceptualization, Investigation, Visualization, Methodology, Writing – original draft, Writing – review and editing; Johannes Allweier, Investigation, Methodology, Writing – review and editing; Joëlle Paolo Mesén-Ramírez, Jan Stäcker, Anna Viktoria Vaaben, Isabel Naranjo-Prado, Susann Ofori, Pascal WTC Jansen, Joëlle Hornebeck, Florian Kieferle, Agnes Murk, Elicia Martin, Carolina Castro-Peña, Methodology, Writing – review and editing; Gala Ramón-Zamorano, Investigation, Writing – review and editing; Max Graser, Patricia López-Barona, Methodology; Richárd Bártfai, Thomas Lavstsen, Supervision, Project administration, Writing – review and editing; Iris Bruchhaus, Conceptualization, Supervision, Funding acquisition, Project administration, Writing – review and editing; Tobias Spielmann, Conceptualization, Supervision, Funding acquisition, Investigation, Visualization, Writing – original draft, Project administration, Writing – review and editing

## Author ORCIDs

Jakob Cronshagen ![ORCID] https://orcid.org/0000-0002-4879-8053
Johannes Allweier ![ORCID] http://orcid.org/0000-0002-5845-5804
Max Graser ![ORCID] http://orcid.org/0009-0003-1990-4712
Patricia López-Barona ![ORCID] http://orcid.org/0000-0002-2691-5263
Carolina Castro-Peña ![ORCID] http://orcid.org/0000-0002-6878-4951
Thomas Lavstsen ![ORCID] https://orcid.org/0000-0002-3044-4249
Iris Bruchhaus ![ORCID] https://orcid.org/0000-0002-3363-7409
Tobias Spielmann ![ORCID] https://orcid.org/0000-0002-3968-4601

Reviewer #1 (Public review): https://doi.org/10.7554/eLife.103542.3.sa1
Reviewer #2 (Public review): https://doi.org/10.7554/eLife.103542.3.sa2
Reviewer #3 (Public review): https://doi.org/10.7554/eLife.103542.3.sa3
Author response https://doi.org/10.7554/eLife.103542.3.sa4

# Additional files

## Supplementary files

Supplementary file 1. Sequences of Plasmid constructs.

MDAR checklist

Source data 1. RNA Seq and differential gene expression results for the tested SLI lines.

Source data 2. Whole genome sequencing comparison of Var01-EMPIC3-TGD and Var01-TryThrA-TGD parasites with the Var01 parent cell line (IT4var01-BirA*Pos1).

## Data availability

All data generated or analyzed during this study are included in the manuscript and supporting files. Source data files have been provided for Figures 1-7. The RNA-Seq data have been deposited in NCBI's Gene Expression Omnibus (REF 103 in the manuscript) and are accessible through GEO Series accession number GSE267413, the whole genome sequencing data with the accession number GSE275671. The mass spectrometry proteomics data have been deposited to the ProteomeXchange Consortium via the PRIDE (REF 104 in the manuscript) partner repository with the dataset identifier PXD052297.

The following datasets were generated:

| Author(s) | Year | Dataset title | Dataset URL | Database and Identifier |
|---|---|---|---|---|
| Cronshagen J, Allweier J, Mesén-Ramírez P, Stäcker J, Viktoria Vaaben A, Ramón-Zamorano G, Naranjo I, Ofori S, Jansen PW, Hornebeck J, Kieferle F, Martin E, Murk A, Castro-Peña C, Bártfai R, Lavstsen T, Bruchhaus I, Spielmann T | 2024 | RNASeq results of SLI lines | https://www.ncbi.nlm.nih.gov/geo/query/acc.cgi?acc=GSE267413 | NCBI Gene Expression Omnibus, GSE267413 |
| Cronshagen J, Allweier J, Mesén-Ramírez P, Stäcker J, Viktoria Vaaben A, Ramón-Zamorano G, Naranjo I, Ofori S, Jansen P WTC, Hornebeck J, Kieferle F, Murk A, Castro-Peña C, Bártfai R, Lavstsen T, Bruchhaus I, Spielmann T | 2024 | Whole genome sequencing samples of erythrocytes infected with the *P. falciparum* transgenic parasites IT4var01-BirA*Pos1endo, IT4var01-BirA*Pos1endo + TryThrA-TGD or IT4var01-BirA*Pos1endo + EMPIC3-TGD | https://www.ncbi.nlm.nih.gov/geo/query/acc.cgi?acc=GSE275671 | NCBI Gene Expression Omnibus, GSE275671 |
| Cronshagen J, Allweier J, Mesén-Ramírez JP, Stäcker J, Vaaben AV, Ramón-Zamorano G, Naranjo-Prado I, Graser M, López-Barona P, Ofori S, Jansen PWTC, Hornebeck J, Kieferle F, Murk A, Martin E, Castro-Peña C, Bártfai R, Lavstsen T, Bruchhaus I, Spielmann T | 2024 | *Plasmodium falciparum* PfEMP1 BioID data | https://www.ebi.ac.uk/pride/archive/projects/PXD052297/ | PRIDE, PXD052297 |

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
