## [Editor Report · eLife Assessment]

This study introduces an **important** approach using selection linked integration (SLI) to generate *Plasmodium falciparum* lines expressing single, specific surface adhesins PfEMP1 variants, enabling precise study of PfEMP1 trafficking, receptor binding, and cytoadhesion. By moving the system to different parasite strains and introducing an advanced SLI2 system for additional genomic edits, this work provides **compelling** evidence for an innovative and rigorous platform to explore PfEMP1 biology and identify novel proteins essential for malaria pathogenesis including immune evasion.

---

## [Referee Report · Reviewer #1 (Public review)]

One of the roadblocks in PfEMP1 research has been the challenges in manipulating var genes to incorporate markers to allow the transport of this protein to be tracked and to investigate the interactions taking place within the infected erythrocyte. In addition, the ability of *Plasmodium falciparum* to switch to different PfEMP1 variants during in vitro culture has complicated studies due to parasite populations drifting from the original (manipulated) var gene expression. Cronshagen et al have provided a useful system with which they demonstrate the ability to integrate a selectable drug marker into several different var genes that allows the PfEMP1 variant expression to be 'fixed'. This on its own represents a useful addition to the molecular toolbox and the range of var genes that have been modified suggests that the system will have broad application. As well as incorporating a selectable marker, the authors have also used selective linked integration (SLI) to introduce markers to track the transport of PfEMP1, investigate the route of transport and probe interactions with PfEMP1 proteins in the infected host cell.

One of the major strengths of this paper is that the authors have not only put together a robust system for further functional studies, but they have used it to produce a range of interesting findings including:

Co-activation of rif and var genes when in a head-to-head orientation.

The reduced control of expression of var genes in the 3D7-MEED parasite line.

More support for the PTEX transport route for PfEMP1.

Identification of new proteins involved in PfEMP1 interactions in the infected erythrocyte, including some required for cytoadherence.

In most cases the experimental evidence is straightforward, and the data support the conclusions strongly. The authors have been very careful in the depth of their investigation, and where unexpected results have been obtained, they have looked carefully at why these have occurred.

A weakness of the paper is, as mentioned above, that the results are sometimes not as clear as might have been expected, for example, in the requirement for panning modified parasites to produce binding to EPCR. Where this has happened, the authors take a robust and thoughtful approach, and acknowledge that (as in most research) there are more questions to address. Being able to select specific var gene switches using drug markers will provide some useful starting points to understand how switching happens in *P. falciparum*. However, our trypanosome colleagues might remind us that forcing switches may show us some mechanisms, but perhaps not all.

Despite these sometimes complicated findings, the authors have achieved their aim as stated in the title of the paper, and in doing so have provided an excellent resource to themselves and other researchers in the field to answer some important questions.

Overall, the authors have produced a useful and robust system to support functional studies on PfEMP1, which provides a platform for future studies manipulating the domain content in var genes. They have used this system to produce a range of interesting findings and to support its use by the research community.

Comments on revisions:

I have no further recommendations for changes by the authors. They have addressed my concerns, and the paper reads very well.

---

## [Referee Report · Reviewer #2 (Public review)]

Summary

Croshagen et al develop a range of tools based on selection-linked integration (SLI) to study PfEMP1 function in *P. falciparum*. PfEMP1 is encoded by a family of ~60 var genes subject to mutually exclusive expression. Switching expression between different family members can modify the binding properties of the infected erythrocyte while avoiding the adaptive immune response. Although critical to parasite survival and Malaria disease pathology, PfEMP1 proteins are difficult to study owing to their large size and variable expression between parasites within the same population. The SLI approach previously developed by this group for genetic modification of *P. falciparum* is employed here to selectively and stably activate expression of target var genes at the population level. Using this strategy, the binding properties of specific PfEMP1 variants were measured for several distinct var genes with a novel semi-automated pipeline to increase throughput and reduce bias. Activation of similar var genes in both the common lab strain 3D7 and the cytoadhesion competent FCR3/IT4 strain revealed higher binding for several PfEMP1 IT4 variants with distinct receptors, indicating this strain provides a superior background for studying PfEMP1 binding. SLI also enables modifications to target var gene products to study PfEMP1 trafficking and identify interacting partners by proximity-labeling proteomics, revealing two novel exported proteins required for cytoadherence. Overall, the data demonstrate a range of SLI-based approaches for studying PfEMP1 that will be broadly useful for understanding the basis for cytoadhesion and parasite virulence.

Comments:

While the capability of SLI to active selected var gene expression was initially reported by Omelianczyk et al., the present study greatly expands the utility of this approach. Several distinct var genes are activated in two different *P. falciparum* strains and shown to modify the binding properties of infected RBCs to distinct endothelial receptors; development of SLI2 enables multiple SLI modifications in the same parasite line; SLI is used to modify target var genes to study PfEMP1 trafficking and determine PfEMP1 interactomes with BioID. Along the way, the authors also demonstrate a new selection marker for *P. falciparum* transfection (a mutant FNT lactate transporter that provides resistance to the compound BH267.meta). Curiously, Omelianczyk et al activated a single var (Pf3D7_0421300) and observed elevated expression of an adjacent var arranged in a head to tail manner, possibly resulting from local chromatin modifications enabling expression of the neighboring gene. In contrast, the present study observed activation of neighboring genes with head to head but not head to tail arrangement, which may be the result of shared promoter regions. The reason for these differing results is unclear although it should be noted that the two studies examined different var loci.

The IT4var19 panned line that became binding-competent showed increased expression of both paralogs of ptp3 (as well as a phista and gbp), suggesting that overexpression of PTP3 may improve PfEMP1 display and binding. Interestingly, IT4 appears to be the only known *P. falciparum* strain (only available in PlasmoDB) that encodes more than one ptp3 gene (PfIT_140083100 and PfIT_140084700). PfIT_140084700 is almost identical to the 3D7 PTP3 (except for a ~120 residue insertion in 3D7 beginning at residue 400). In contrast, while the C-terminal region of PfIT_140083100 shows near perfect conservation with 3D7 PTP3 beginning at residue 450, the N-terminal regions between the PEXEL and residue 450 are quite different. This may indicate the generally stronger receptor binding observed in IT4 relative to 3D7 results from increased PTP3 activity due to multiple isoforms or that specialized trafficking machinery exists for some PfEMP1 proteins.

Revisions:

The authors thoughtfully addressed all the reviewer comments.

---

## [Referee Report · Reviewer #3 (Public review)]

Summary:

The submission from Cronshagen and colleagues describes the application of a previously described method (selection linked integration) to the systematic study of PfEMP1 trafficking in the human malaria parasite *Plasmodium falciparum*. PfEMP1 is the primary virulence factor and surface antigen of infected red blood cells and is therefore a major focus of research into malaria pathogenesis. Since the discovery of the var gene family that encodes PfEMP1 in the late 1990s, there have been multiple hypotheses for how the protein is trafficked to the infected cell surface, crossing multiple membranes along the way. One difficulty in studying this process is the large size of the var gene family and the propensity of the parasites to switch which var gene is expressed, thus preventing straightforward gene modification-based strategies for tagging the expressed PfEMP1. Here the authors solve this problem by forcing expression of a targeted var gene by fusing the PfEMP1 coding region with a drug selectable marker separated by a skip peptide. This enabled them to generate relatively homogenous populations of parasites all expressing tagged (or otherwise modified) forms of PfEMP1 suitable for study. They then applied this method to study various aspects of PfEMP1 trafficking.

Strengths:

The study is very thorough, and the data are well presented. The authors used SLI to target multiple var genes, thus demonstrating the robustness of their strategy. They then perform experiments to investigate possible trafficking through PTEX, they knockout proteins thought to be involved in PfEMP1 trafficking and observe defects in cytoadherence, and they perform proximity labeling to further identify proteins potentially involved in PfEMP1 export. These are independent and complimentary approaches that together tell a very compelling story.

Weaknesses:

(1) When the authors targeted IT4var19, they were successful in transcriptionally activating the gene, however they did not initially obtain cytoadherent parasites. To observe binding to ICAM-1 and EPCR, they had to perform selection using panning. This is an interesting observation and potentially provides insights into PfEMP1 surface display, folding, etc. However, it also raises questions about other instances in which cytoadherence was not observed. Would panning of these other lines have successfully selected for cytoadherent infected cells? Did the authors attempt panning of their 3D7 lines? Given that these parasites do export PfEMP1 to the infected cell surface (Figure 1D), it is possible that panning would similarly rescue binding. Likewise, the authors knocked out PTP1, TryThrA and EMPIC3 and detected a loss of cytoadhesion, but they did not attempt panning to see if this could rescue binding. The strong selection that panning exerts on parasite populations could result in selection of compensatory changes that enable cytoadherence, which could be very informative, although the analysis could potentially be quite complicated and beyond the scope of the current paper. Nonetheless, these are important concepts to consider when assessing these phenotypes.

(2) The authors perform a series of trafficking experiments to help discern whether PfEMP1 is trafficked through PTEX. While the results were not entirely definitive, they make a strong case for PTEX in PfEMP1 export. The authors then used BioID to obtain a proxiome for PfEMP1 and identified proteins they suggest are involved in PfEMP1 trafficking. However, it seemed that components of PTEX were missing from the list of interacting proteins. Is this surprising and does this observation shed any additional light on the possibility of PfEMP1 trafficking through PTEX? This warrants a comment or discussion.

Comments on revisions:

The authors have responded thoroughly and constructively to suggestions and comments in the initial review. I have no additional comments. This is a great contribution to the literature.

---

## [Author Response]

The following is the authors’ response to the original reviews.

**eLife Assessment:**
This study introduces an important approach using selection linked integration (SLI) to generate *Plasmodium falciparum* lines expressing single, specific surface adhesins PfEMP1 variants, enabling precise study of PfEMP1 trafficking, receptor binding, and cytoadhesion. By moving the system to different parasite strains and introducing an advanced SLI2 system for additional genomic edits, this work provides compelling evidence for an innovative and rigorous platform to explore PfEMP1 biology and identify novel proteins essential for malaria pathogenesis including immune evasion.
**Reviewer #1 (Public review):**
One of the roadblocks in PfEMP1 research has been the challenges in manipulating var genes to incorporate markers to allow the transport of this protein to be tracked and to investigate the interactions taking place within the infected erythrocyte. In addition, the ability of *Plasmodium falciparum* to switch to different PfEMP1 variants during in vitro culture has complicated studies due to parasite populations drifting from the original (manipulated) var gene expression. Cronshagen et al have provided a useful system with which they demonstrate the ability to integrate a selectable drug marker into several different var genes that allows the PfEMP1 variant expression to be 'fixed'. This on its own represents a useful addition to the molecular toolbox and the range of var genes that have been modified suggests that the system will have broad application. As well as incorporating a selectable marker, the authors have also used selective linked integration (SLI) to introduce markers to track the transport of PfEMP1, investigate the route of transport, and probe interactions with PfEMP1 proteins in the infected host cell.What I particularly like about this paper is that the authors have not only put together what appears to be a largely robust system for further functional studies, but they have used it to produce a range of interesting findings including:Co-activation of rif and var genes when in a head-to-head orientation.The reduced control of expression of var genes in the 3D7-MEED parasite line.More support for the PTEX transport route for PfEMP1.Identification of new proteins involved in PfEMP1 interactions in the infected erythrocyte, including some required for cytoadherence.In most cases the experimental evidence is straightforward, and the data support the conclusions strongly. The authors have been very careful in the depth of their investigation, and where unexpected results have been obtained, they have looked carefully at why these have occurred.

We thank the reviewer for the kind assessment and the comments to improve the paper.

(1) In terms of incorporating a drug marker to drive mono-variant expression, the authors show that they can manipulate a range of var genes in two parasite lines (3D7 and IT4), producing around 90% expression of the targeted PfEMP1. Removal of drug selection produces the expected 'drift' in variant types being expressed. The exceptions to this are the 3D7-MEED line, which looks to be an interesting starting point to understand why this variant appears to have impaired mutually exclusive var gene expression and the EPCR-binding IT4var19 line. This latter finding was unexpected and the modified construct required several rounds of panning to produce parasites expressing the targeted PfEMP1 and bind to EPCR. The authors identified a PTP3 deficiency as the cause of the lack of PfEMP1 expression, which is an interesting finding in itself but potentially worrying for future studies. What was not clear was whether the selected IT4var19 line retained specific PfEMP1 expression once receptor panning was removed.

We do not have systematic long-term data for the Var19 line but do have medium-term data. After panning the Var19 line, the binding assays were done within 3 months without additional panning. The first binding assay was 2 months after the panning and the last binding assays three weeks later, totaling about 3 months without panning. While there is inherent variation in these assays that precludes detection of smaller changes, the last assay showed the highest level of binding, giving no indication for rapid loss of the binding phenotype. Hence, we can say that the binding phenotype appears to be stable for many weeks without panning the cells again and there was no indication for a rapid loss of binding in these parasites.

Systematic long-term experiments to assess how long the Var19 parasites retain binding would be interesting, but given that the binding-phenotype appears to remain stable over many weeks or even months, this would only make sense if done over a much longer time frame. Such data might arise if the line is used over extended times for a specific project in which case it might be advisable to monitor continued binding. We included a statement in the discussion that the binding phenotype was stable over many weeks but that if long-term work with this line is planned, monitoring the binding phenotype might be advisable: “In the course of this work the binding phenotype of the IT4var19 expressor line remained stable over many weeks without further panning. However, given that initial panning had been needed for this particular line, it might be advisable for future studies to monitor the binding phenotype if the line is used for experiments requiring extended periods of cultivation.”

(2) The transport studies using the mDHFR constructs were quite complicated to understand but were explained very clearly in the text with good logical reasoning.

We are aware of this being a complex issue and are glad this was nevertheless understandable.

(3) By introducing a second SLI system, the authors have been able to alter other genes thought to be involved in PfEMP1 biology, particularly transport. An example of this is the inactivation of PTP1, which causes a loss of binding to CD36 and ICAM-1. It would have been helpful to have more insight into the interpretation of the IFAs as the anti-SBP1 staining in Figure 5D (PTP-TGD) looks similar to that shown in Figure 1C, which has PTP intact. The anti-EXP2 results are clearly different.

We realize the description of the PTP1-TGD IFA data and that of the other TGDs (see also response to Recommendation to authors point 4 and reviewer 2, major points 6 and 7) was rather cursory. The previously reported PTP1 phenotype is a fragmentation of the Maurer’s clefts into what in IFA appear to be many smaller pieces (Rug et al 2014, referenced in the manuscript). The control in Fig. 5D has 13 Maurer’s cleft spots (previous work indicates an average of ~15 MC per parasite, see e.g. the originally co-submitted eLife preprint doi.org/10.7554/eLife.103633.1 and references therein). The control mentioned by the reviewer in Fig. 1C has about 22 Maurer’s clefts foci, at the upper end of the typical range, but not unusual. In contrast, the PTP1-TGD in Fig. 5D, has more than 30 foci with an additional cytoplasmic pool and additional smaller, difficult to count foci. This is consistent with the published phenotype in Rug et al 2014. The EXP1 stained cell has more than 40 Maurer’s cleft foci, again beyond what typically is observed in controls. Therefore, these cells show a difference to the control in Fig. 5 but also to Fig. 1C. Please note that we are looking at two different strains, in Fig. 1 it is 3D7 and in Fig. 5 IT4. While we did not systematically assess this, the Maurer’s clefts number per cell seemed to be largely comparable between these strains (Fig. 10C and D in the other eLife preprint doi.org/10.7554/eLife.103633.1).

Overall, as the PTP1 loss phenotype has already been reported, we did not go into more experimental detail. However, we now modified the text to more clearly describe how the phenotype in the PTP1-TGD parasites was different to control: “IFAs showed that in the PTP1-TGD parasites, SBP1 and PfEMP1 were found in many small foci in the host cell that exceeded the average number of ~ 15 Maurer’s clefts typically found per infected RBC [66] (Fig. 5D). This phenotype resembled the previously reported Maurer’s clefts phenotype of the PTP1 knock out in CS2 parasites [39].”

(4) It is good to see the validation of PfEMP1 expression includes binding to several relevant receptors. The data presented use CHO-GFP as a negative control, which is relevant, but it would have been good to also see the use of receptor mAbs to indicate specific adhesion patterns. The CHO system if fine for expression validation studies, but due to the high levels of receptor expression on these cells, moving to the use of microvascular endothelial cells would be advisable. This may explain the unexpected ICAM-1 binding seen with the panned IT4var19 line.

We agree with the reviewer that it is desirable to have better binding systems for studying individual binding interactions. As the main purpose of this paper was to introduce the system and provide proof of principle that the cells show binding, we did not move to more complicated binding systems. However, we would like to point out that the CSA binding was done on receptor alone in addition to the CSA-expressing HBEC-5i cells and was competed successfully with soluble CSA. In addition, apart from the additional ICAM1-binding of the Var19 line, all binding phenotypes were conform with expectations. We therefore hope the tools used for binding studies are acceptable at this stage of introducing the system while future work interested in specific PfEMP1 receptor interactions may use better systems, tailored to the specific question (e.g. endothelial organoid models and engineered human capillaries and inhibitory antibodies or relevant recombinant domains for competition).

(5) The proxiome work is very interesting and has identified new leads for proteins interacting with PfEMP1, as well as suggesting that KAHRP is not one of these. The reduced expression seen with BirA* in position 3 is a little concerning but there appears to be sufficient expression to allow interactions to be identified with this construct. The quantitative impact of reduced expression for proxiome experiments will clearly require further work to define it.

This is a valid point. Clearly there seems to be some impact on binding when BirA* is placed in the extracellular domain (either through reduced presentation or direct reduction of binding efficiency of the modified PfEMP1; please see also minor comment 10 reviewer 2). The exact quantitative impact on the proxiome is difficult to assess but we note that the relative enrichment of hits to each other is rather similar to the other two positions (Fig. 6H-J). We therefore believe the BioIDs with the 3 PfEMP1-BirA* constructs are sufficient to provide a general coverage of proteins proximal to PfEMP1 and hope this will aid in the identification of further proteins involved in PfEMP1 transport and surface display as illustrated with two of the hits targeted here.

The impact of placing a domain on the extracellular region of PfEMP1 will have to be further evaluated if needed in other studies. But the finding that a large folded domain can be placed into this part at all, even if binding was reduced, in our opinion is a success (it was not foreseeable whether any such change would be tolerated at all).

(6) The reduced receptor binding results from the TryThrA and EMPIC3 knockouts were very interesting, particularly as both still display PfEMP1 on the surface of the infected erythrocyte. While care needs to be taken in cross-referencing adhesion work in P. berghei and whether the machinery truly is functionally orthologous, it is a fair point to make in the discussion. The suggestion that interacting proteins may influence the "correct presentation of PfEMP1" is intriguing and I look forward to further work on this.

We hope future work will be able to shed light on this.

Overall, the authors have produced a useful and reasonably robust system to support functional studies on PfEMP1, which may provide a platform for future studies manipulating the domain content in the exon 1 portion of var genes. They have used this system to produce a range of interesting findings and to support its use by the research community. Finally, a small concern. Being able to select specific var gene switches using drug markers could provide some useful starting points to understand how switching happens in *P. falciparum*. However, our trypanosome colleagues might remind us that forcing switches may show us some mechanisms but perhaps not all.

Point noted! From non-systematic data with the Var01 line that has been cultured for extended periods of time (several years), it seems other non-targeted vars remain silent in our SLI “activation” lines but how much SLI-based var-expression “fixing” tampers with the integrity of natural switching mechanisms is indeed very difficult to gage at this stage. We now added a statement to the discussion that even if mutually exclusive expression is maintained, it is not certain the mechanisms controlling var expression all remain intact: “However, it should be noted that it is not known whether all mechanisms controlling mutually exclusive expression and switching remain intact in parasites with SLI-activated var genes.”

**Reviewer #2 (Public review):**
SummaryCroshagen et al develop a range of tools based on selection-linked integration (SLI) to study PfEMP1 function in *P. falciparum*. PfEMP1 is encoded by a family of ~60 var genes subject to mutually exclusive expression. Switching expression between different family members can modify the binding properties of the infected erythrocyte while avoiding the adaptive immune response. Although critical to parasite survival and Malaria disease pathology, PfEMP1 proteins are difficult to study owing to their large size and variable expression between parasites within the same population. The SLI approach previously developed by this group for genetic modification of *P. falciparum* is employed here to selectively and stably activate the expression of target var genes at the population level. Using this strategy, the binding properties of specific PfEMP1 variants were measured for several distinct var genes with a novel semi-automated pipeline to increase throughput and reduce bias. Activation of similar var genes in both the common lab strain 3D7 and the cytoadhesion competent FCR3/IT4 strain revealed higher binding for several PfEMP1 IT4 variants with distinct receptors, indicating this strain provides a superior background for studying PfEMP1 binding. SLI also enables modifications to target var gene products to study PfEMP1 trafficking and identify interacting partners by proximity-labeling proteomics, revealing two novel exported proteins required for cytoadherence. Overall, the data demonstrate a range of SLI-based approaches for studying PfEMP1 that will be broadly useful for understanding the basis for cytoadhesion and parasite virulence.

We thank the reviewer for the kind assessment and the comments to improve the paper.

Comments(1) While the capability of SLI to actively select var gene expression was initially reported by Omelianczyk et al., the present study greatly expands the utility of this approach. Several distinct var genes are activated in two different *P. falciparum* strains and shown to modify the binding properties of infected RBCs to distinct endothelial receptors; development of SLI2 enables multiple SLI modifications in the same parasite line; SLI is used to modify target var genes to study PfEMP1 trafficking and determine PfEMP1 interactomes with BioID. Curiously, Omelianczyk et al activated a single var (Pf3D7_0421300) and observed elevated expression of an adjacent var arranged in a head-to-tail manner, possibly resulting from local chromatin modifications enabling expression of the neighboring gene. In contrast, the present study observed activation of neighboring genes with head-to-head but not head-totail arrangement, which may be the result of shared promoter regions. The reason for these differing results is unclear although it should be noted that the two studies examined different var loci.

The point that we are looking at different loci is very valid and we realize this is not mentioned in the discussion. We now added to the discussion that it is unclear if our results and those cited may be generalized and that different var gene loci may respond differently

“However, it is unclear if this can be generalized and it is possible that different var loci respond differently.”

(2) The IT4var19 panned line that became binding-competent showed increased expression of both paralogs of ptp3 (as well as a phista and gbp), suggesting that overexpression of PTP3 may improve PfEMP1 display and binding. Interestingly, IT4 appears to be the only known *P. falciparum* strain (only available in PlasmoDB) that encodes more than one ptp3 gene (PfIT_140083100 and PfIT_140084700). PfIT_140084700 is almost identical to the 3D7 PTP3 (except for a ~120 residue insertion in 3D7 beginning at residue 400). In contrast, while the C-terminal region of PfIT_140083100 shows near-perfect conservation with 3D7 PTP3 beginning at residue 450, the N-terminal regions between the PEXEL and residue 450 are quite different. This may indicate the generally stronger receptor binding observed in IT4 relative to 3D7 results from increased PTP3 activity due to multiple isoforms or that specialized trafficking machinery exists for some PfEMP1 proteins.

We thank the reviewer for pointing this out, the exact differences between the two PTP3s of IT4 and that of other strains definitely should be closely examined if the function of these proteins in PfEMP1 binding is analysed in more detail.

It is an interesting idea that the PTP3 duplication could be a reason for the superior binding of IT4. We always assumed that IT4 had better binding because it was less culture adapted but this does not preclude that PTP3(s) is(are) a reason for this. However, at least in our 3D7 PTP3 can’t be the reason for the poor binding, as our 3D7 still has PfEMP1 on the surface while in the unpanned IT4-Var19 line and in the Maier et al., Cell 2008 ptp3 KO (PMID: 18614010) PfEMP1 is not on the surface anymore.

Testing the impact of having two PTP3s would be interesting, but given the “mosaic” similarity of the two PTP3s isoforms, a simple add-on experiment might not be informative. Nevertheless, it will be interesting in future work to explore this in more detail.

**Reviewer #3 (Public review):**
Summary:The submission from Cronshagen and colleagues describes the application of a previously described method (selection linked integration) to the systematic study of PfEMP1 trafficking in the human malaria parasite *Plasmodium falciparum*. PfEMP1 is the primary virulence factor and surface antigen of infected red blood cells and is therefore a major focus of research into malaria pathogenesis. Since the discovery of the var gene family that encodes PfEMP1 in the late 1990s, there have been multiple hypotheses for how the protein is trafficked to the infected cell surface, crossing multiple membranes along the way. One difficulty in studying this process is the large size of the var gene family and the propensity of the parasites to switch which var gene is expressed, thus preventing straightforward gene modification-based strategies for tagging the expressed PfEMP1. Here the authors solve this problem by forcing the expression of a targeted var gene by fusing the PfEMP1 coding region with a drug-selectable marker separated by a skip peptide. This enabled them to generate relatively homogenous populations of parasites all expressing tagged (or otherwise modified) forms of PfEMP1 suitable for study. They then applied this method to study various aspects of PfEMP1 trafficking.Strengths:The study is very thorough, and the data are well presented. The authors used SLI to target multiple var genes, thus demonstrating the robustness of their strategy. They then perform experiments to investigate possible trafficking through PTEX, they knock out proteins thought to be involved in PfEMP1 trafficking and observe defects in cytoadherence, and they perform proximity labeling to further identify proteins potentially involved in PfEMP1 export. These are independent and complimentary approaches that together tell a very compelling story.

We thank the reviewer for the kind assessment and the comments to improve the paper.

Weaknesses:(1) When the authors targeted IT4var19, they were successful in transcriptionally activating the gene, however, they did not initially obtain cytoadherent parasites. To observe binding to ICAM-1 and EPCR, they had to perform selection using panning. This is an interesting observation and potentially provides insights into PfEMP1 surface display, folding, etc. However, it also raises questions about other instances in which cytoadherence was not observed. Would panning of these other lines have been successfully selected for cytoadherent infected cells? Did the authors attempt panning of their 3D7 lines? Given that these parasites do export PfEMP1 to the infected cell surface (Figure 1D), it is possible that panning would similarly rescue binding. Likewise, the authors knocked out PTP1, TryThrA, and EMPIC3 and detected a loss of cytoadhesion, but they did not attempt panning to see if this could rescue binding. To ensure that the lack of cytoadhesion in these cases is not serendipitous (as it was when they activated IT4var19), they should demonstrate that panning cannot rescue binding.

These are very important considerations. Indeed, we had repeatedly attempted to pan 3D7 when we failed to get the SLI-generated 3D7 PfEMP1 expressor lines to bind, but this had not been successful. The lack of binding had been a major obstacle that had held up the project and was only solved when we moved to IT4 which readily bound (apart from Var19 which was created later in the project). After that we made no further efforts to understand why 3D7 does not bind but the fact that PfEMP1 is on the surface indicates this is not a PTP3 issue because loss of PTP3 also leads to loss of PfEMP1 surface display. Also, as the parent 3D7 could not be panned, we assumed this issue is not easily fixed in the SLI var lines we made in 3D7.

Panning the TGD lines: we see the reasoning for conducting panning experiments with the TGD lines. However, on second thought, we are unsure this should be attempted. The outcome might not be easily interpretable as at least two forces will contribute to the selection in panning experiments with TGD lines that do not bind anymore:

Firstly, panning would work against the SLI of the TGD, resulting in a tug of war between the TGD-SLI and binding. This is because a small number of parasites will loop out the TGD plasmid (revert) and would normally be eliminated during standard culturing due to the SLI drug used for the TGD. These revertant cells would bind and the panning would enrich them. Hence, panning and SLI are opposed forces in the case of a TGD abolishing binding. It is unclear how strong this effect would be, but this would for sure lead to mixed populations that complicate interpretations.

The second selecting force are possible compensatory changes to restore binding. These can be due to different causes: (i) reversal of potential independent changes that may have occurred in the TGD parasites and that are in reality causing the binding loss (i.e. such as ptp3 loss or similar, the concern of the reviewer) or (ii) new changes to compensate the loss of the TGD target (in this case the TGD is the cause of the binding loss but for instance a different change ameliorates it by for instance increasing PfEMP1 expression or surface display). As both TGDs show some residual binding and have VAR01 on the surface to at least some extent, it is possible that new compensatory changes might indeed occur that indirectly increase binding again.

In summary, even if more binding occurs after panning of the lines, it is not clear whether this is due to a compensatory change ameliorating the TGD or reversal of an unrelated change or are counter-selections against the SLI. To determine the cause, the panned TGD lines would need to be subjected to a complex and time-consuming analysis (WGS, RNASeq, possibly Maurer’s clefts phenotype) to find out whether they were SLI-revertants, or had an unrelated chance that was reverted or a new compensatory change that helps binding. This might be further muddled if a mix of cells come out of the selection that have different changes of the options indicated above. In that case, it might even require scRNASeq to make sense of the panning experiment. Due to the envisaged difficulty in interpreting the outcome, we did not attempt this panning.

To exclude loss of ptp3 expression as the reason for binding loss (something we would not have seen in the WGS if it is only due to a transcriptional change), we now carried out RNASeq with the TGD lines that have a binding phenotype. While we did not generate replicas to obtain quantitative data, the results show that both ptp3 copies were expressed in these TGDs comparable to other parasite lines that do bind with the same SLI-activated var gene, indicating that the effect is not due to ptp3 (see response to point 4 on PTP3 expression in the Recommendations for the authors). While we can’t fully exclude other changes in the TGDs that might affect binding, the WGS did not show any obvious alterations that could be responsible for this.

(2) The authors perform a series of trafficking experiments to help discern whether PfEMP1 is trafficked through PTEX. While the results were not entirely definitive, they make a strong case for PTEX in PfEMP1 export. The authors then used BioID to obtain a proxiome for PfEMP1 and identified proteins they suggest are involved in PfEMP1 trafficking. However, it seemed that components of PTEX were missing from the list of interacting proteins. Is this surprising and does this observation shed any additional light on the possibility of PfEMP1 trafficking through PTEX? This warrants a comment or discussion.

This is an interesting point and we agree that this warrants to be discussed. A likely reason why PTEX components are not picked up as interactors is that BirA* is expected to be unfolded when it passes through the channel and in that state can’t biotinylate. Labelling likely would only be possible if PfEMP1 lingered at the PTEX translocation step before BirA* became unfolded to go through the channel which we would not expect under physiological conditions. We added the following sentences to the discussion: “While our data indicates PfEMP1 uses PTEX to reach the host cell, this could be expected to have resulted in the identification of PTEX components in the PfEMP1 proxiomes, which was not the case. However, as BirA* must be unfolded to pass through PTEX, it likely is unable to biotinylate translocon components unless PfEMP1 is stalled during translocation. For this reason, a lack of PTEX components in the PfEMP1 proxiomes does not necessarily exclude passage through PTEX.”

**Recommendations for the authors:**

**Reviewer #1 (Recommendations for the authors):**
Most of my comments are in the public section. I would just highlight a few things:(1) In the binding studies section you talk about "human brain endothelial cells (HBEC-5i)". These cells do indeed express CSA but this is a property of their immortalisation rather than being brain endotheliium, which does not express CSA. I think this could be confusing to readers so I think you might want to reword this sentence to focus on CSA expressing the cell line rather than other features.

We thank the reviewer for pointing this out, we now modified the sentence to focus on the fact these are CSA expressing cells and provided a reference for it.

(2) As I said in the public section, CHO cells are great for proof of concept studies, but they are not endothelium. Not a problem for this paper.

Noted! Please also see our response to the public review.

(3) I wonder whether your comment about how well tolerated the Bir3* insertion is may be a bit too strong. I might say "Nonetheless, overall the BirA* modified PfEMP1 were functional."

Changed as requested.

(4) I'm not sure how you explain the IFA staining patterns to the uninitiated, but perhaps you could explain some of the key features you are looking for.

We apologise for not giving an explanation of the IFA staining patterns in the first place. Please see detailed response to public review of this reviewer (point 3 on PTP1-TGD phenotype) and to reviewer 2 (Recommendations to the authors, points 6 and 7 on better explaining and quantifying the Maurer’s clefts phenotypes). For this we now also generated parasites that episomally express mCherry tagged SBP1 in the TGD parasites with the reduced binding phenotype. This resulted in amendments to Fig. S7, addition of a Fig. S8 and updated results to better explain the phenotypes.

This is a great paper - I just wish I'd had this system before.

Thank you!

**Reviewer #2 (Recommendations for the authors):**
Major Comments(1) Does the RNAseq analysis of 3D7var0425800 and 3D7MEEDvar0425800 (Figure 1G, H) reveal any differential gene expression that might suggest a basis for loss of mutually exclusive var expression in the MEED line?

We now carried out a thorough analysis of these RNASeq experiments to look for an underlying cause for the phenotype. This was added as new Figure 1J and new Table S3. This analysis again illustrated the increased transcript levels of var genes. In addition, it showed that transcripts of a number of other exported proteins, including members of other gene families, were up in the MEED line.

One hit that might be causal of the phenotype was sip2, which was down by close to 8-fold (pAdj 0.025). While recent work in P. berghei found this ApiAP2 to be involved in the expression of merozoite genes (Nishi et al., Sci Advances 2025(PMID: 40117352)), previous work in *P. falciparum* showed that it binds heterochromatic telomere regions and certain var upstream regions (Flück et al., PlosPath 2010 (PMID: 20195509), now cited in the manuscript). The other notable change was an upregulation of the non-coding RNA ruf6 which had been linked with impaired mono-allelic var expression (Guizetti et al., NAR 2016 (PMID: 27466391), now also cited in the manuscript). While it would go beyond this manuscript to follow this up, it is conceivable that alterations in chromosome end biology due to sip2 downregulation or upregulation of ruf6 are causes of the observed phenotype

We now added a paragraph on the more comprehensive analysis of the RNA Seq data of the MEED vs non-MEED lines at the end of the second results section.

(2) Could the inability of the PfEMP1-mDHFR fusion to block translocation (Fig 2A) reflect unique features of PfEMP1 trafficking, such as the existence of a soluble, chaperoned trafficking state that is not fully folded? Was a PfEMP1-BPTI fusion ever tested as an alternative to mDHFR?

This is an interesting suggestion. The PfEMP1-BPTI was never tested. However, a chaperoned trafficking state would likely also affect BPTI. Given that both domains (mDHFR and BPTI) in principle do the same when folded and would block when the construct is in the PV, it is not so likely that using a different blocking domain would make a difference. Therefore, the scenario where BPTI would block when mDHFR does not, is not that probable. The opposite would be possible (mDHFR blocking while BPTI does not, because only the latter depends on the redox state). However, this would only happen if the block occurred before the construct reaches the PV.

At present, we believe the lacking block to be due to the organization of the domains in the construct. In the PfEMP1-mDHFR construct in this manuscript the position of the blocking domain is further away from the TMD compared to all other previously tested mDHFR fusions. Increased distance to the TMD has previously been found to be a factor impairing the blocking function of mDHFR (Mesen-Ramirez et al., PlosPath 2016 (PMID: 27168322)). Hence, our suspicion that this is the reason for the lacking block with the PfEMP1-mDHFR rather than the type of blocking domain. However, the latter option can’t be fully excluded and we might test BPTI in future work.

(3) The late promoter SBP1-mDHFR is 2A fused with the KAHRP reporter. Since 2A skipping efficiency varies between fusion contexts and significant amounts of unskipped protein can be present, it would be helpful to include a WB to determine the efficiency of skipping and provide confidence that the co-blocked KAHRP in the +WR condition (Fig 2D) is not actually fused to the C-terminus of SBP1-mDHFR-GFP.

Fortunately, this T2A fusion (crt_SBP1-mDHFR-GFP-2A-KAHRP-mScarlet^epi^) was used before in work that included a Western blot showing its efficient skipping (S3 A Fig in MesenRamirez et al., PlosPath 2016). In agreement with these Western blot result, fluorescence microscopy showed very limited overlap of SBP1-mDHFR-GFP and KAHRP-mCherry in absence of WR (Fig. 3B in Mesen-Ramirez et al., PlosPath 2016 and Fig. 2 in this manuscript) which would not be the case if these two constructs were fused together. Please note that KAHRP is known to transiently localize to the Maurer’s clefts before reaching the knobs (Wickham et al., EMBOJ 2001, PMID: 11598007), and therefore occasional overlap with SBP1 at the Maurer’s clefts is expected. However, we would expect much more overlap if a substantial proportion of the construct population would not be skipped and therefore the co-blocked KAHRP-mCherry in the +WR sample is unlikely to be due to inefficient skipping and attachment to SBP1-mDHFR-GFP.

(4) Does comparison of RNAseq from the various 3D7 and IT4 lines in the study provide any insight into PTP3 expression levels between strains with different binding capacities? Was the expression level of ptp3a/b in the IT4var19 panned line similar to the expression in the parent or other activated IT4 lines? Could the expanded ptp3 gene number in IT4 indicate that specialized trafficking machinery exists for some PfEMP1 proteins (ie, IT4var19 requires the divergent PTP3 paralog for efficient trafficking)?

PTP3 in the different IT4 lines that bind:

In those parasite lines that did bind, the intrinsic variation in the binding assays, the different binding properties of different PfEMP1 variants and the variation in RNA Seq experiments to compare different parasite lines precludes a correlation of binding level vs ptp3 expression. For instance, if a PfEMP1 variant has lower binding capacity, ptp3 may still be higher but binding would be lower than if comparing to a parasite line with a better binding PfEMP1 variant. Studying the effect of PTP3 levels on binding could probably be done by overexpressing PTP3 in the same PfEMP1 SLI expressor line and assessing how this affects binding, but this would go beyond this manuscript.

PTP3 in panned vs unpanned Var19:

We did some comparisons between IT4 parent, and the IT4-Var19 panned and unpanned

(see Author response table 1). This did not reveal any clear associations. While the parent had somewhat lower ptp3 transcript levels, they were still clearly higher than in the unpanned Var19 line and other lines had also ptp3 levels comparable to the panned IT4-Var19 (see Author response table 2)

PTP3 in the TGDs and possible reason for binding phenotype:

A key point is whether PTP3 could have influenced the lack of binding in the TGD lines (see also weakness section and point 1 of public review of reviewer 3: ptp3 may be an indirect cause resulting in lacking binding in TGD parasites). We now did RNA Seq to check for ptp3 expression in the relevant TGD lines although we did not do a systematic quantitative comparison (which would require 3 replicates of RNASeq), but we reasoned that loss of expression would also be evident in one replicate. There was no indication that the TGD lines had lost PTP3 expression (see Author response table 2) and this is unlikely to explain the binding loss in a similar fashion to the Var19 parasites. Generally, the IT4 lines showed expression of both ptp3 genes and only in the Var19 parasites before panning were the transcript levels considerably lower:

**Author response table 1. sa4table1:** Parent vs IT4-Var19 panned and unpanned.

Gene ID	Average of IT4 parent (n=3)	Average Var19 panned (n=4)	Average Var19 not panned (n=4)	
PfiT_140083100	2,45596041	3,14231686	0,35897933	
PfiT _140084700	4,97115609	17,5952948	0,84662599	

**Author response table 2. sa4table2:** TGD lines with binding phenotype vs parent.

Gene ID	SLI_BioIDPos1_Empic3 -TGD	SLI_BioIDPos1_TryThraA -TGD	SLI_ptp1-TGD	SLI_BirAPos1_var
PfiT_140083100	2,25443223	1,06975218	1,20352604	1,4791184
PfiT_140084700	3,41126305	22,3717269	18,7216101	14,2297336

The absence of an influence of PTP3 on the binding phenotype in the cell lines in this manuscript (besides Var19) is further supported by its role in PfEMP1 surface display. Previous work has shown that KO of ptp3 leads to a loss of VAR2CSA surface display (Maier et al., Cell 2008). The unpanned Var19 parasite also lacked PfEMP1 surface display and panning and the resulting appearance of the binding phenotype was accompanied by surface display of PfEMP1. As both, the EMPIC3 and TryThra-TGD lines had still at least some PfEMP1 on the surface, this also (in addition to the RNA Seq above) speaks against PTP3 being the cause of the binding phenotype. The same applies to 3D7 which despite the poor binding displays PfEMP1 on the host cell surface (Figure 1D). This indicating that also the binding phenotype in 3D7 is not due to PTP3 expression loss, as this would have abolished PfEMP1 surface display.

The idea about PTP3 paralogs for specific PfEMP1s is intriguing. In the future it might be interesting to test the frequency of parasites with two PTP3 paralogs in endemic settings and correlate it with the PfEMP1 repertoire, variant expression and potentially disease severity.

(5) The IT4var01 line shows substantially lower binding in Figure 5F compared with the data shown in Figure 4E and 6F. Does this reflect changes in the binding capacity of the line over time or is this variability inherent to the assay?

There is some inherent variability in these assays. While we did not systematically assess this, we had no indication that this was due to the parasite line changing. The Var01 line was cultured for months and was frozen down and thawed more than once without a clear gradual trend for more or less binding. While we can’t exclude some variation from the parasite side, we suspect it is more a factor of the expression of the receptor on the CHO cells the iRBCs bind to.

Specifically, the assays in Fig. 6F and 4E mentioned by the reviewer both had an average binding to CD36 of around 1000 iE/mm2, only the experiments in Fig. 5F are different (~ 500 iE/mm2) but these were done with a different batch of CHO cells at a different time to the experiments in Fig. 6F and 4E.

(6) In Figure S7A, TryThrA and EMPIC3 show distinct localization as circles around the PfEMP1 signal while PeMP2 appears to co-localize with PfEMP1 or as immediately adjacent spots (strong colocalization is less apparent than SBP1, and the various PfEMP1 IFAs throughout the study). Does this indicate that TryThrA and EMPIC3 are peripheral MC proteins? Does this have any implications for their function in PfEMP1 binding? Some discussion would help as these differences are not mentioned in the text. For the EMPIC3 TGD IFAs, localization of SBP1 and PfEMP1 is noted to be normal but REX1 is not mentioned (although this also appears normal).

We apologise for the lacking description of the candidate localisations and cursory description of the Maurer’s clefts phenotypes (next point). Our original intent was to not distract too much from the main flow of the manuscript as almost every part of the manuscript could be followed up with more details. However, we fully agree that this is unsatisfactory and now provided more description (this point) and more data (next point).

Localisation of TryThrA and EMPIC3 compared to PfEMP1 at the Maurer’s clefts: the circular pattern is reminiscent of the results with Maurer’s clefts proteins reported by McMillan et al using 3D-SIM in 3D7 parasites (McMillan et al., Cell Microbiology 2014 (PMID: 23421990)). In that work SBP1 and MAHRP1 (both integral TMD proteins) were found in foci but REX1 (no TMD) in circular structures around these foci similar to what we observed here for TryThrA and EMPIC3 which both also lack a TMD. The SIM data in McMillan et al indicated that also PfEMP1 is “more peripheral”, although it did only partially overlap with REX1. The conclusion from that work was that there are sub-compartments at the Maurer’s clefts. In our IFAs (Fig. S7A) PfEMP1 is also only partially overlapping with the TryThrA and EMPIC3 circles, potentially indicating similar subcompartments to those observed by 3D-SIM. We agree with the reviewer that this might be indicative of peripheral MC proteins, fitting with a lack of TMD in these candidates, but we did not further speculate on this in the manuscript.

We now added enlargements of the ring-like structures to better illustrate this observation in Fig. S7A. In addition, we now specifically mention the localization data and the ring like signal with TryThrA and EMPIC3 in the results and state that this may be similar to the observations by McMillan et al., Cell Microbiology 2014.

We also thank the reviewer for pointing out that we had forgotten to mention REX1 in the EMPIC3-TGD, this was amended.

(7) The atypical localization in TryThrA TGD line claimed for PfEMP1 and SBP1 in Fig S7B is not obvious. While most REX1 is clustered into a few spots in the IFA staining for SBP1 and REX1, SBP1 is only partially located in these spots and appears normal in the above IFA staining for SBP1 and HA. The atypical localization of PfEMP1-HA is also not obvious to me. The authors should clarify what is meant by "atypical" localization and provide support with quantification given the difference between the two SBP1 images shown.

We apologise for the inadequate description of these IFA phenotypes. The abnormal signal for SBP1, REX1 and PfEMP1 in the TryThrA-TGD included two phenotypes found with all 3 proteins:

(1) a dispersed signal for these proteins in the host cell in addition to foci (the control and the other TGD parasites have only dots in the host cell with no or very little detectable dispersed signal).

(2) foci of disproportionally high intensity and size, that we assumed might be aggregation or enlargement of the Maurer’s clefts or of the detected proteins.

The reason for the difference between the REX1 (aggregation) phenotype and the PfEMP1 and SBP1 (dispersed signal, more smaller foci) phenotypes in the images in Fig. S7B is that both phenotypes were seen with all 3 proteins but we chose a REX1 stained cell to illustrate the aggregation phenotype (the SBP1 signal in the same cell is similar to the REX1 signal, illustrating that this phenotype is not REX1 specific; please note that this cell also has a dispersed pool of REX1 and SBP1).

Based on the IFAs 66% (n = 106 cells) of the cells in the TryThrA-TGD parasites had one or both of the observed phenotypes. We did not include this into the previous version of the manuscript because a description would have required detouring from the main focus of this results section. In addition, IFAs have some limitations for accurate quantifications, particularly for soluble pools (depending on fixing efficiency and agent, more or less of a soluble pool in the host cell can leak out).

To answer the request to better explain and quantify the phenotype and given the limitations of IFA, we now transfected the TryThrA-TGD parasites with a plasmid mediating episomal expression of SBP1-mCherry, permitting live cell imaging and a better classification of the Maurer’s clefts phenotype. Due to the two SLI modifications in these parasites (using up 4 resistance markers) we had to use a new selection marker (mutated lactate transporter PfFNT, providing resistance to BH267.meta (Walloch et al., J. Med. Chem. 2020 (PMID: 32816478))) to transfect these parasites with an additional plasmid.

These results are now provided as Fig. S8 and detailed in the last results section. The new data shows that the majority of the TryThrA-TGD parasites contain a dispersed pool of SBP1 in the host cell. About a third of the parasites also showed disproportionally strong SBP1 foci that may be aggregates of the Maurer’s clefts. We also transfected the EMPIC3-TGD parasites with the FNT plasmid mediating episomal SBP1-mCherry expression and observed only few cells with a cytoplasmic pool or aggregates (Fig. S8). Overall these findings agree with the previous IFA results. As the IFA suggests similar results also for REX1 and PfEMP1, this defect is likely not SBP1 specific but more general (Maurer’s clefts morphology; association or transport of multiple proteins to the Maurer’s clefts). This gives a likely explanation for the cytoadherence phenotype in the TryThrA-TGD parasites. The reason for the EMPIC3-TGD phenotype remains to be determined as we did not detect obvious changes of the Maurer’s clefts morphology or in the transport of proteins to these structures in these experiments.

Minor comments(1) Italicized numbers in parenthesis are present in several places in the manuscript but it is not clear what these refer to (perhaps differently formatted citations from a previous version of the manuscript). Figure 1legend: (121); Figure S3 legend: (110), (111); Figure S6 legend: (66); etc.

We thank the reviewer for pointing out this issue with the references, this was amended.

(2) Figure 5A and legend: "BSD-R: BSD-resistance gene". Blasticidin-S (BS) is the drug while Blasticidin-S deaminase (BSD) is the resistance gene.

We thank the reviewer for pointing this out, the legend and figure were changed.

(3) Figure 5E legend: µ-SBP1-N should be α-SBP1-N.

This was amended.

(4) Figure S5 legend: "(Full data in Table S1)" should be Table S3.

This was amended.

(5) Figure S1G: The pie chart shows PF3D7_0425700 accounts for 43% of rif expression in 3D7var0425800 but the text indicates 62%.

We apologize for this mistake, the text was corrected. We also improved the citations to Fig. S1G and H in this section.

(6) "most PfEMP1-trafficking proteins show a similar early expression..." The authors might consider including a table of proteins known to be required for EMP1 trafficking and a graph showing their expression timing. Are any with later expressions known?

Most exported proteins are expressed early, which is nicely shown in Marti et al 2004 (cited for the statement) in a graph of the expression timing of all PEXEL proteins (Fig. 4B in that paper). PNEPs also have a similar profile (Grüring et al 2011, also cited for that statement), further illustrated by using early expression as a criterion to find more PNEPs (Heiber et al., 2013 (PMID: 23950716)). Together this includes most if not all of the known PfEMP1 trafficking proteins. The originally co-submitted paper (Blancke-Soares & Stäcker et al., eLife preprint doi.org/10.7554/eLife.103633.1) analysed several later expressed exported proteins

(Pf332, MSRP6) but their disruption, while influencing Maurer’s clefs morphology and anchoring, did not influence PfEMP1 transport. However, there are some conflicting results for Pf332 (referenced in Blancke-Soares & Stäcker et al). This illustrates that it may not be so easy to decide which proteins are bona fide PfEMP1 trafficking proteins. We therefore did not add a table and hope it is acceptable for the reader to rely on the provided 3 references to back this statement.

(7) Figure S1J: The predominate var in the IT4 WT parent is var66 (which appears to be syntenic with Pf3D7_0809100, the predominate var in the 3D7 WT parent). Is there something about this locus or parasite culture conditions that selects for these vars in culture? Is this observed in other labs as well?

This is a very interesting point (although we are not certain these vars are indeed syntenic, they are on different chromosomes). As far as we know at least Pf3D7_0809100 is commonly a dominant var transcribed in other labs and was found expressed also in sporozoites (Zanghì et al. Cell Rep. 2018). However, it is unclear how uniform this really is. For IT4 we do not know in full but have also here commonly observed centromeric var genes to be dominating transcripts in unselected parasite cultures. It is possible that transcription drifts to centromeric var genes in cultured parasites. However, given the anecdotal evidence, it is unknown to which extent this is related to an inherent switching and regulation regiment or a consequence of faulty regulation following prolonged culturing.

(8) Figure 4B, C: Presumably the asterisks on the DNA gels indicate non-specific bands but this is not described in the legend. Why are non-specific bands not consistent between parent and integrated lanes?

We apologize for not mentioning this in the legend, this was amended.

It is not clear why the non-specific bands differ between the lines but in part this might be due to different concentrations and quality of DNA preps. A PCR can also behave differently depending on whether the correct primer target is present or not. If present, the PCR will run efficiently and other spurious products will be outcompeted, but in absence of the correct target, they might become detectable.

Overall, we do not think the non-specific bands are indications of anything untoward with the lines, as for instance in Fig. 4B the high band in the 5’ integration in the IT4 line (that does not occur anywhere else) can’t be due to a genomic change as this is the parental line and does not contain the plasmid for integration. In the same gel, the ori locus band of incorrect size (likely due to crossreaction of the primers to another var gene which due to the high similarity of the ATS region is not always fully avoidable), is present in both, the parent IT4 and the integrant line which therefore also is not of concern. In C there are a couple of bands of incorrect size in the Integration line. One of these is very faint and both are too large and again therefore are likely other vars that are inefficiently picked up by these primers. The reason they are not seen in the parent line is that there the correct primer binding site is present, which then efficiently produces a product that outcompetes the product derived from non-optimal matching primer products and hence appear in the Int line where the correct match is not there anymore. For these reasons we believe these bands are not of any concern.

(9) Figure 4C: Is there a reason KAHRP was used as a co-marker for the IFA detecting IT4var19 expression instead of SBP1 which was used throughout the rest of the study?

This is a coincidence as this line was tested when other lines were tested for KAHRP. As there were foci in the host cell we were satisfied that the HA-tagged PfEMP1 is produced and the localization deemed plausible.

(10) Figure 6: Streptavidin labeling for the IT4var01-BirA position 3 line is substantially less than the other two lines in both IFA and WB. Does the position 3 fusion reduce PfEMP1 protein levels or is this a result of the context or surface display of the fusion? Interestingly, the position 3 trypsin cleavage product appears consistently more robust compared with the other two configurations. Does this indicate that positioning BirA upstream of the TM increases RBC membrane insertion and/or makes the surface localized protein more accessible to trypsin?

It is possible that RBC membrane insertion or trypsin accessibility is increased for the position 3 construct. But there could also be other explanations:

The reason for the more robustly detected protected fragment for the position 3 construct in the WB might also be its smaller size (in contrast to the other two versions, it does not contain BirA*) which might permit more efficient transfer to the WB membrane. In that case the more robust band might not (only) be due to better membrane insertion or better trypsin accessibility.

The lower biotinylation signal with the position 3 construct might also be explained by the farther distance of BirA* to the ATS (compared to position 1 and 2), the region where interactors are expected to bind. The position 1 and 2 constructs may therefore generally be more efficient (as closer) to biotinylate ATS proximal proteins. Further, in the final destination (PfEMP1 inserted into the RBC membrane) BirA* would be on the other side of the membrane in the position 3 construct while in the position 1 and 2 constructs BirA* would be on the side of the membrane where the ATS anchors PfEMP1 in the knob structure. In that case, labelling with position 3 would come from interactions/proximities during transport or at the Maurer’s clefts (if there indeed PfEMP1 is not membrane embedded) and might therefore be less.

Hence, while alterations in trypsin accessibility and RBC membrane insertion are possible explanations, other explanations exist. At present, we do not know which of these explanations apply and therefore did not mention any of them in the manuscript.

**Reviewer #3 (Recommendations for the authors):**
(1) In the abstract and on page 8, the authors mention that they generate cell lines binding to "all major endothelial receptors" and "all known major receptors". This is a pretty allencompassing statement that might not be fully accepted by others who have reported binding to other receptors not considered in this paper (e.g. VCAM, TSP, hyaluronic acid, etc). It would be better to change this statement to something like "the most common endothelial receptors" or "the dominant endothelial receptors", or something similar.

We agree with the reviewer that these statements are too all-encompassing and changed them to “the most common endothelial receptors” (introduction) and “the most common receptors” (results).

(2) The authors targeted two rif genes for activation and in each case the gene became the most highly expressed member of the family. However, unlike var genes, there were other rif genes also expressed in these lines and the activated copy did not always make up the majority of rif mRNAs. The authors might wish to highlight that this is inconsistent with mutually exclusive expression of this gene family, something that has been discussed in the past but not definitively shown.

We thank the reviewer for highlighting this, we now added the following statement to this section: “While SLI-activation of rif genes also led to the dominant expression of the targeted rif gene, other rif genes still took up a substantial proportion of all detected rif transcripts, speaking against a mutually exclusive expression in the manner seen with var genes.”

(3) In Figure 6, H-J, the authors display volcano plots showing proteins that are thought to interact with PfEMP1. These are labeled with names from the literature, however, several are named simply "1, 2, 3, 4, 5, or 6". What do these numbers stand for?

We apologize for not clarifying this and thank the reviewer for pointing this out. There is a legend for the numbered proteins in what is now Table S4 (previously Table S3). We now amended the legend of Figure 6 to explain the numbers and pointing the reader to Table S4 for the accessions.